# Quantum control and Berry phase of electron spins in rotating levitated diamonds in high vacuum

Yuanbin Jin [1,7], Kunhong Shen[1,7], Peng Ju[1], Xingyu Gao[1], Chong Zu [2], Alejandro J. Grine[3] & Tongcang Li [1,4,5,6] ✉

Levitated diamond particles in high vacuum with internal spin qubits have been proposed for exploring macroscopic quantum mechanics, quantum gravity, and precision measurements. The coupling between spins and particle rotation can be utilized to study quantum geometric phase, create gyroscopes and rotational matter-wave interferometers. However, previous efforts in levitated diamonds struggled with vacuum level or spin state readouts. To address these gaps, we fabricate an integrated surface ion trap with multiple stabilization electrodes. This facilitates on-chip levitation and, for the first time, optically detected magnetic resonance measurements of a nanodiamond levitated in high vacuum. The internal temperature of our levitated nanodiamond remains moderate at pressures below $10^{-5}$ Torr. We have driven a nanodiamond to rotate up to 20 MHz ($1.2 \times 10^9$ rpm), surpassing typical nitrogen-vacancy (NV) center electron spin dephasing rates. Using these NV spins, we observe the effect of the Berry phase arising from particle rotation. In addition, we demonstrate quantum control of spins in a rotating nanodiamond. These results mark an important development in interfacing mechanical rotation with spin qubits, expanding our capacity to study quantum phenomena.

Levitated nanoparticles and microparticles in high vacuum[1–3] offer a remarkable degree of isolation from environmental noises, rendering them exceptionally suitable for studying fundamental physics[4–6] and executing precision measurements[7–11]. Recently, the center-of-mass (CoM) motion of levitated nanoparticles in high vacuum has been cooled to the quantum regime[12–14]. Unlike tethered oscillators, levitated particles can also exhibit rotation[15–22], which is intrinsically nonlinear[23,24]. Beyond rigid-body motion, levitated particles can host embedded spin qubits to provide more functionalities[25,26]. Notably, levitated nanodiamonds with embedded NV center spin qubits have been proposed for creating massive quantum superpositions[25,26] to

test the limit of quantum mechanics and quantum gravity[27,28]. The embedded spin qubits can also sense the pseudo-magnetic field[29–31], related to the Barnett effect[32,33], and the quantum geometric phase[34,35] associated with particle rotation. The coupling between spin and mechanical rotation can be utilized for building sensitive gyroscopes[36,37] and rotational matter-wave interferometers[38,39]. These innovative proposals necessitate levitating diamond particles in high vacuum, well below $10^{-3}$ Torr. However, prior experiments with levitated diamonds struggled with vacuum level or spin state readouts.

Optical levitation of nanodiamonds has been experimentally achieved, but it was restricted to pressures above 1 Torr due to

[1]Department of Physics and Astronomy, Purdue University, West Lafayette, IN 47907, USA. [2]Department of Physics, Washington University, St. Louis, MO 63130, USA. [3]Sandia National Laboratories, Albuquerque, NM 87185, USA. [4]Elmore Family School of Electrical and Computer Engineering, Purdue University, West Lafayette, IN 47907, USA. [5]Purdue Quantum Science and Engineering Institute, Purdue University, West Lafayette, IN 47907, USA. [6]Birck Nanotechnology Center, Purdue University, West Lafayette, IN 47907, USA. [7]These authors contributed equally: Yuanbin Jin, Kunhong Shen. ✉e-mail: tcli@purdue.edu

laser-induced heating[40–42]. Earlier studies using Paul traps, or ion traps, have demonstrated spin cooling[43] and angle locking[44] of levitated diamonds. However, they encountered a similar issue: diamond particles were lost when the air pressure was reduced to about 0.01 Torr[43–47]. This phenomenon could be due to the nonideal design of the Paul traps used in those experiments or the heating effects of detection lasers. While nanodiamonds can be levitated in a magneto-gravitational trap[48,49], reading out the spin state within this setup remains elusive.

In this article, we design and fabricate an integrated surface ion trap (Fig. 1a) that incorporates an Ω-shaped stripline to deliver both a low-frequency high voltage for trapping and a microwave for NV spin control. Additionally, it comprises multiple electrodes to stabilize the trap and drive a levitated diamond to rotate. With this advanced Paul trap, we have performed optically detected magnetic resonance (ODMR) measurements of a levitated nanodiamond in high vacuum for the first time. Using NV spins, we measure the internal temperature of the levitated nanodiamond, which remains stable at approximately 350 K under pressures below $10^{-5}$ Torr. This suggests prospects for levitation in ultra-high vacuum. With a rotating electric field, we have been able to drive a levitated nanodiamond to rotate at high speeds up to 20 MHz ($1.2 \times 10^9$ rpm), which is about three orders of magnitudes faster than previous achievements using diamonds mounted on motor spindles[29,30]. Notably, this rotation speed surpasses the typical dephasing rate of NV spins in the diamond. With embedded NV electron spins in the levitated nanodiamond, we observe the effect of the Berry phase generated by the mechanical rotation, which also improves the ODMR spectrum in an external magnetic field. Moreover, we achieve quantum control of NV centers in a rotating levitated nanodiamond. Our work represents a pivotal advancement in interfacing mechanical rotation with spin qubits.

## Results

### Levitation of a nanodiamond in high vacuum

In the experiment, we levitate a nanodiamond in vacuum using a surface ion trap (Fig. 1a). The surface ion trap is fabricated on a sapphire wafer, which has high transmittance for visible and near-infrared lasers. To achieve levitation of nanodiamonds and quantum control of NV spins simultaneously, we apply both an AC high voltage and a microwave on a Ω-shaped circuit. The AC high voltage has a frequency of about 20 kHz and an amplitude of about 200 V. The microwave has a frequency of a few GHz. They are combined together with a home-made bias tee. The center ring electrode is grounded to generate a trapping center above the chip surface. The four electrodes at the corners are used to compensate the static electric fields from surface charges to minimize the micro-motion of a levitated nanodiamond. Figure 1c shows a simulated distribution of the electric field of the trap. The trapping center is 253 $\mu$m away from the chip surface.

The trapping potential depends on the charge-to-mass ratio ($Q/m$) of a levitated particle. Thus, it is necessary to increase the charge number of particles for stable levitation in an ion trap. In our experiment, nanodiamonds are charged and sprayed out by electrospray and delivered to the surface ion trap with an extra linear Paul trap. The charge of the sprayed nanodiamond is typically larger than 1000 $e$, where $e$ is the elementary charge, enabling a large trapping depth of more than 100 eV (see Supplementary Note 1 for more details). A 532 nm laser is used to excite diamond NV centers and a 1064 nm laser is

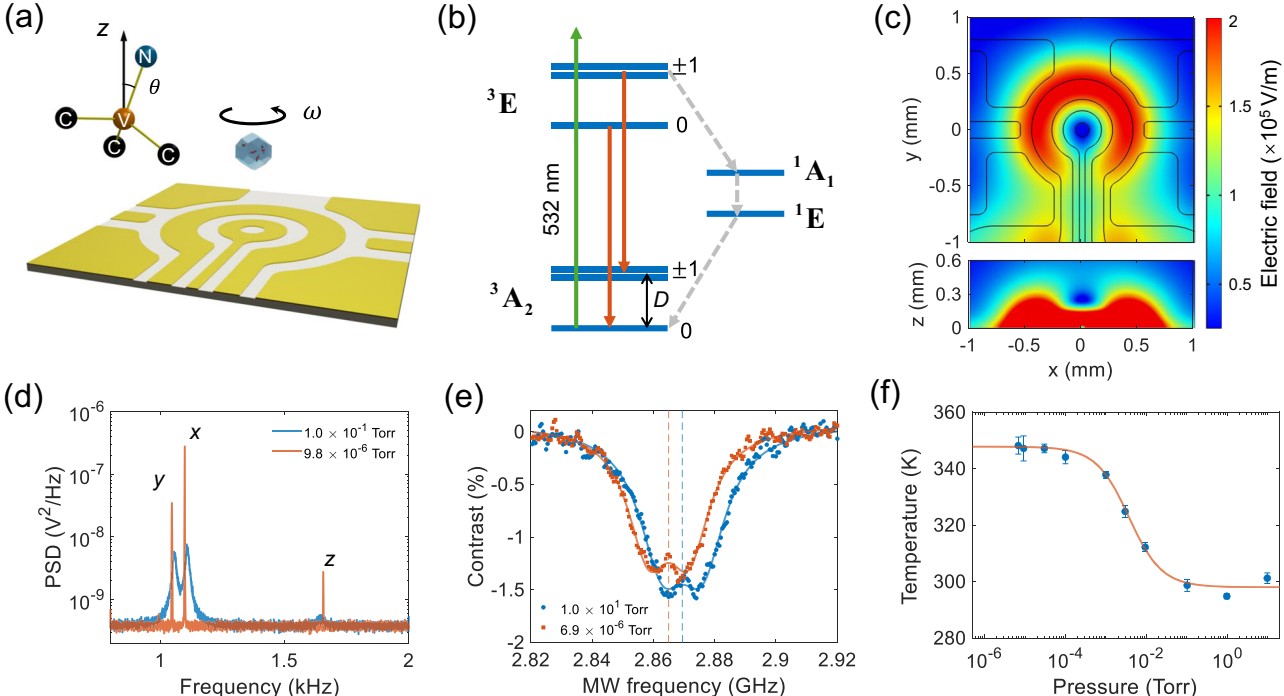

**Fig. 1 | Stable levitation of a nanodiamond in high vacuum. a** Schematic of a levitated nanodiamond in a surface ion trap. The center ring electrode is grounded (GND). It has a hole at its center for sending a 1064 nm laser to monitor the nanodiamond's motion. A combination of a low-frequency high voltage (HV) and a high-frequency microwave (MW) is applied to the Ω-shaped circuit to trap the nanodiamond and control the NV centers. **b** Energy level diagram of a diamond NV center. A 532 nm laser (green arrow) excites the NV center. The red solid arrows and gray dashed arrows represent radiative decays and nonradiative decays, respectively. **c** Simulation of the electric field of the ion trap in the *xy*-plane (top) and in the *xz*-plane (bottom) when a voltage of 200 V is applied to the Ω-shaped circuit. The trap center is 253 $\mu$m away from the chip surface. **d** Power spectrum densities (PSDs) of the center-of-mass (CoM) motion of the levitated nanodiamond at the pressure of 0.1 Torr (blue) and $9.8 \times 10^{-6}$ Torr (red). **e** Optically detected magnetic resonances (ODMRs) of the levitated nanodiamond measured at 10 Torr (blue circles) and $6.9 \times 10^{-6}$ Torr (red squares). The blue and red dashed lines are the corresponding zero-field splittings. The intensities of the 532 nm laser and the 1064 nm laser are 0.030 W/mm² and 0.520 W/mm², respectively. **f** Internal temperature of the levitated nanodiamond as a function of pressure with the same laser intensities as shown in (e). Error bars represent the standard deviation of temperature among three measurements.

(a) 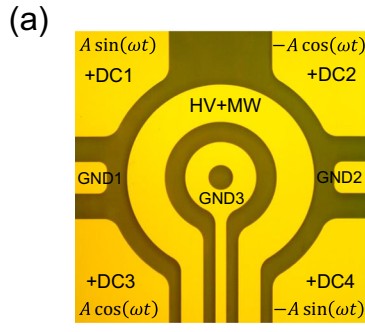

(b) 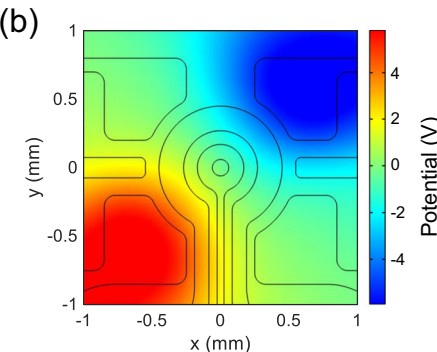

(c) 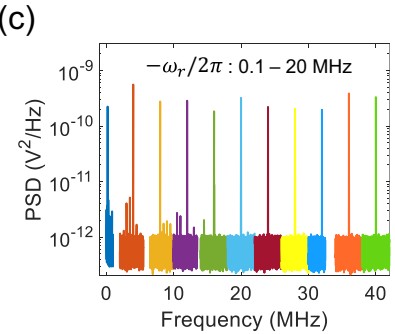

**Fig. 2 | Fast rotation of a levitated nanodiamond. a** Optical image of the surface ion trap. AC voltage signals ($A\sin(\omega t + \varphi)$) with the same frequency ($\omega$) and amplitude ($A$) but different phases ($\varphi$) are applied to the four corner electrodes to generate a rotating electric field. The phase is different by $\pi/2$ between neighboring electrodes. DC1, DC2, DC3 and DC4 are compensation voltages that minimize the micromotion to stabilize the trap. **b** Simulation of the electric potential in the $z = 253\,\mu m$ plane at $t = 0$. The amplitude is $A = 10\,V$. (c) PSDs of the rotational motion of the levitated nanodiamond at the rotation frequencies from 0.1 MHz to 20 MHz. The pressure is $1.0 \times 10^{-4}$ Torr.

applied to monitor the nanodiamond's motion. More details of our experimental setup are shown in Supplementary Fig. 1.

A main result of our experiment is that we can levitate a nanodiamond with the surface ion trap in high vacuum, which is a breakthrough as levitated diamond particles were lost around 0.01 Torr in previous studies using ion traps[43–47]. The red curve in Fig. 1d shows the power spectrum density (PSD) of the center-of-mass (CoM) motion of a levitated nanodiamond at $9.8 \times 10^{-6}$ Torr. The radius of the levitated nanodiamond is estimated to be about 264 nm based on its PSDs at 0.01 Torr (Supplementary Fig. 2). Our surface ion trap is remarkably stable in high vacuum. We can levitate a nanodiamond in high vacuum continuously for several weeks.

The internal temperature of a levitated nanodiamond is important as it will affect the spin coherence time and trapping stability. We measure the internal temperature using NV centers. The energy levels of an NV center is shown in Fig. 1b. We use a 532 nm laser to excite the NV centers and a single photon counting module to detect their PL. Then we sweep the frequency of a microwave to perform the ODMR measurement of a levitated nanodiamond in the absence of an external magnetic field. Figure 1e shows the ODMRs measured at 10 Torr (blue circles) and $6.9 \times 10^{-6}$ Torr (red squares). Based on the fitting of the ODMRs, the corresponding zero-field splittings (blue and red dashed lines) are 2.8694 GHz and 2.8650 GHz, respectively. The internal temperature of the levitated nanodiamond can be obtained from the zero-field splitting (see Methods and Supplementary Note 2 for details). The measured internal temperature at different pressures are shown in Fig. 1f. The internal temperature is close to the room temperature at pressures above 0.1 Torr, and increases when we reduce the pressure from 0.1 Torr to $10^{-4}$ Torr. Finally, it remains stable at approximately 350 K at pressures below $5 \times 10^{-5}$ Torr. This temperature is low enough to maintain quantum coherence of NV spins for quantum control[50].

The observed phenomena (Fig. 1f) arise from the balance between laser-induced heating (Supplementary Fig. 3) and the cooling effects of air molecules and black-body radiation on the internal temperature of a levitated nanodiamond[51,52]. When the air pressure is high, the cooling rate due to surrounding air molecules is large and the internal temperature of the levitated nanodiamond is close to the room temperature. However, as air pressure decreases, cooling from air molecules diminishes, leading to a rise in internal temperature. When the pressure is below $5 \times 10^{-5}$ Torr, the temperature stabilizes as the cooling is dominated by the black body radiation, which is independent of the air pressure.

**Fast rotation and Berry phase**

After a nanodiamond is levitated in high vacuum, we use a rotating electric field to drive the levitated nanodiamond to rotate at high speeds, which also stabilizes the orientation of the levitated nanodiamond. The four electrodes at the corners are applied with AC voltage signals ($A\sin(\omega t + \varphi)$) with the same frequency ($\omega$) and amplitude ($A$) but different phases ($\varphi$) to generate a rotating electric field (Fig. 2a). The phases of neighboring signals are different by $\pi/2$. Figure 2b shows the simulation of the electric potential in the $xy$-plane at $t = 0$. More information can be found in Supplementary Note 3, Supplementary Fig. 4, and Supplementary Fig. 5. A levitated charged object naturally has an electric dipole moment due to inhomogeneous distribution of charges. In a rotating electric field, the levitated charged particle will rotate due to the torque produced by the interaction between the rotating electric field and the electric dipole of the particle. Figure 2c depicts the PSDs of the rotation at different driving frequencies (0.1–20 MHz). The maximum rotation frequency is 20 MHz in the experiment, which is limited by our phase shifters used to generate phase delays between signals on the four electrodes. This is about 3 orders of magnitudes faster than previous achievements using diamonds mounted on electric motor spindles[29,30]. When the rotation frequency is 100 kHz, the linewidth of the PSD of the rotational signal is fitted to be about $9.9 \times 10^{-5}$ Hz (Supplementary Fig. 5d), which is limited by the measurement time. This shows that the rotation is extremely stable and is locked to the driving electric signal. With easy control and ultra-stability, this driving scheme enables us to adjust and lock the rotation of the levitated nanodiamond over a large range of frequencies (see Supplementary Note 3 for more details).

The fast-rotating diamond with embedded NV spins allows us to observe the effects of the Berry phase due to mechanical rotation. The Berry phase, also known as the geometric phase, is a fundamental aspect of quantum mechanics with applications in multiple fields, including the topological phase of matter and the quantum hall effect[53–57]. The Berry phase in the laboratory frame is equivalent to the pseudo-magnetic field (called the Barnett field in ref. 57) in the rotating frame: $B_\omega = \omega_r/\gamma$, where $\gamma$ is the spin gyromagnetic ratio. In this work, the microwave source is fixed in the laboratory frame. Only the levitated diamond is rotating. So, we can observe the effect of the Berry phase due to rotation[57].

In a rotating diamond, the embedded NV centers also follow the rotation with an angular frequency of $\omega_r$ (Fig. 3). The levitated nanodiamond in our experiment contains ensembles of NV centers with four groups of orientations. Figure 3d shows an NV center embedded in a nanodiamond rotating around the $z$-axis in the presence of an external magnetic field. The direction of the magnetic field is along the $z$-axis. The angle between the NV axis and $z$-axis is $\theta$, and the azimuth is $\phi(t)$ relative to the $x$-axis. The Hamiltonian of the rotating NV electron spin in the laboratory frame, neglecting strain effects, can be

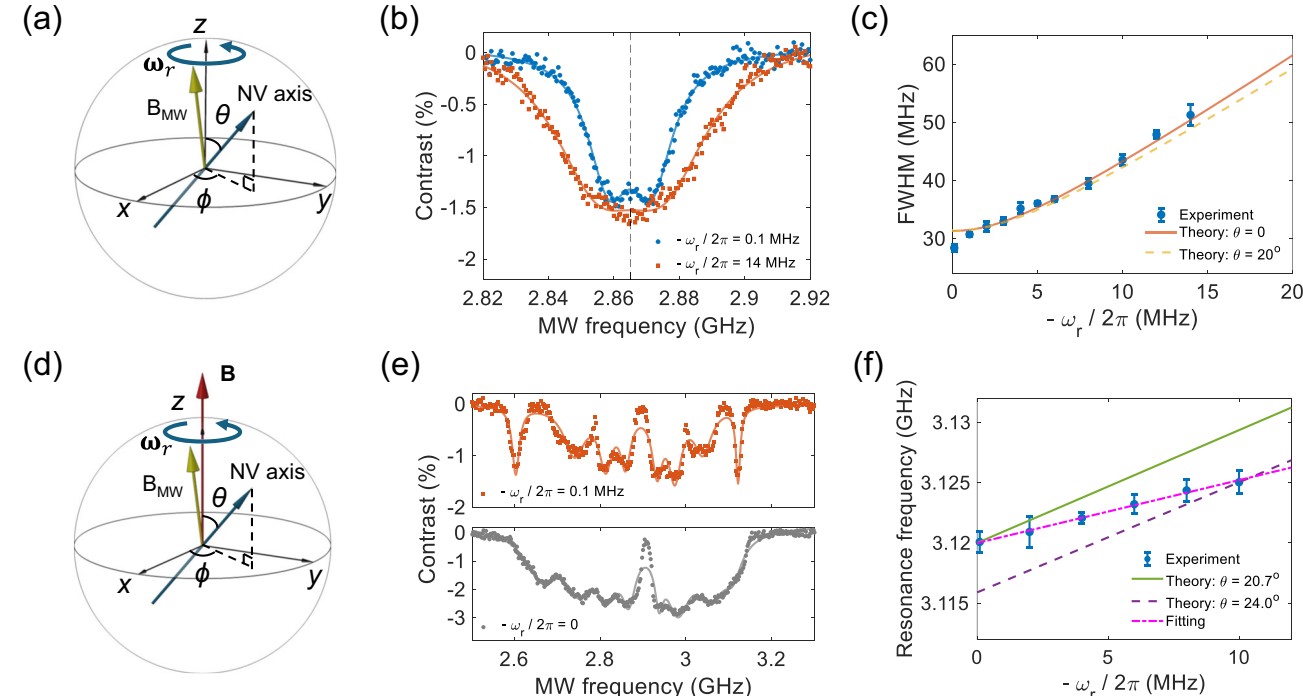

**Fig. 3 | Effects of the Berry phase generated by a rotating nanodiamond.**
**a** Schematic of an NV center in the nanodiamond rotating around the z-axis in the absence of an external magnetic field. The small angle between $B_{MW}$ and the z-axis is due to the asymmetric design of the waveguide. **b** ODMRs of the levitated nanodiamond at rotation frequencies of 0.1 MHz (blue circles) and 14 MHz (red squares). **c** Experimentally measured FWHM of the ODMR spectrum as a function of rotation frequency (blue circles). Error bars show the standard deviations among three measurements. The red solid curve and orange dashed curve are theoretically calculated FWHMs at $\theta = 0°$ and $\theta = 20°$, respectively. **d** Schematic of an NV center in the nanodiamond rotating around the z-axis in an external magnetic field. The magnetic field is along the z-axis and is about 100 G. **e** The upper panel (red

squares) shows the ODMR of the levitated nanodiamond at a rotation frequency of 0.1 MHz and a pressure of $1.0 \times 10^{-4}$ Torr. The bottom panel (gray circles) shows the ODMR of a nanodiamond without a stable rotation at the pressure of 10 Torr. The corresponding solid curves are the fittings with eight Lorentzian dips.
**f** Experimentally measured frequency of the right-most dip of the ODMR spectrum of an NV center as a function of rotation frequency (blue circles). The green solid curve and violet dashed curve are theoretical calculations at $\theta = 20.7°$ and $\theta = 24.0°$, respectively. The magenta dashed curve is a linear fitting of the resonance frequency. The error bars represent standard deviations among three measurements.

---

written as[34]:

$$H_{lab} = H_{0,lab} + g\mu_B B S_z = \frac{1}{\hbar} R(t) D S_z^2 R^\dagger(t) + g\mu_B B S_z$$

$$= D\hbar \begin{pmatrix} \cos^2\theta + \frac{\sin^2\theta}{2} + \frac{g\mu_B B}{D} & \frac{e^{-i\phi}\cos\theta\sin\theta}{\sqrt{2}} & \frac{e^{-2i\phi}\sin^2\theta}{2} \\ \frac{e^{i\phi}\cos\theta\sin\theta}{\sqrt{2}} & \sin^2\theta & -\frac{e^{-i\phi}\cos\theta\sin\theta}{\sqrt{2}} \\ \frac{e^{2i\phi}\sin^2\theta}{2} & -\frac{e^{i\phi}\cos\theta\sin\theta}{\sqrt{2}} & \cos^2\theta + \frac{\sin^2\theta}{2} - \frac{g\mu_B B}{D} \end{pmatrix}, \quad (1)$$

where $D$ is the zero-field splitting, $R(t) = R_z(\phi(t))R_y(\theta)$ is the rotation transformation, and $R_j(\theta) = \exp(-i\theta\mathbf{n}\cdot\mathbf{S})$ for the rotation angle $\theta$ around the $\mathbf{n}$ direction, $j = y, z$, and $\mathbf{S}$ are the spin operators. The Stark shift for NV centers induced by the electric field is negligible and hence is not included in the equation. The Hamiltonian possesses three eigenstates $|m_s, t\rangle_{lab}$ ($m_s = 0, \pm 1$). The detailed expressions can be found in the Supplementary Note 4. Based on its definition, the Berry phase can be calculated as[57]

$$\gamma_{m_s} = i\int_0^t {}_{lab}\langle m_s, t'|\frac{\partial}{\partial t'}|m_s, t'\rangle_{lab}dt' = m_s\omega_r t\cos\theta. \quad (2)$$

Here the Berry phase is calculated for an open-path and is hence gauge-dependent. The spin state of the NV center is observed through the interaction with a microwave magnetic field. In our experiment, the direction of the microwave is in the yz-plane and has a small angle $\theta' = 8.5°$ relative to the z axis, resulting from the asymmetric design of the waveguide. However, the dominant transition probability arises from the longitudinal (z) component. The expected value of the

transition probability of the spin states interacting with the microwave can be expressed as

$$_{lab}\langle \pm 1, t|e^{iH_{lab}t/\hbar}e^{-i\gamma_{\pm 1}}H_{MW,z,lab}e^{i\gamma_0}e^{-iH_{lab}t/\hbar}|0, t\rangle_{lab}$$
$$= \frac{1}{2}g\mu_B B_{MW}\cos\theta' e^{i(-\omega_{MW} + D \pm g\mu_B B\cos\theta \mp \omega_r\cos\theta)t}{}_{lab}\langle \pm 1, 0|e^{i\theta S_y}S_z e^{-i\theta S_y}|0, 0\rangle_{lab}. \quad (3)$$

According to Eq. (3), the transition of spin states from $|m_s = 0\rangle_{lab}$ to $|m_s = \pm 1\rangle_{lab}$ can be driven by a microwave at the resonance frequency of $D \pm g\mu_B B\cos\theta \mp \omega_r\cos\theta$, where the frequency shift $\mp \omega_r\cos\theta$ is due to the Berry phase induced by the mechanical rotation.

We first investigate the effect of the Berry phase in the absence of an external magnetic field. Figure 3a shows the diagram of an NV center rotating around the z-axis without an external magnetic field. To observe the frequency shift due to fast rotation, ODMR measurements of the levitated nanodiamond are carried out at different rotation frequencies. Figure 3b displays ODMRs at the rotation frequencies of 0.1 MHz (bule circles) and 14 MHz (red squares). The full width at half maximum (FWHM) of the ODMR at $\omega_r = 2\pi \times 14$ MHz is clearly larger than that at $\omega_r = 2\pi \times 0.1$ MHz, which is caused by the Berry phase due to rotation. The FWHM of the ODMR at different rotation frequencies is shown in Fig. 3c. The blue circles are the experimental results. The red and orange curves are theoretical results for $\theta = 0°$ and $\theta = 20°$, respectively. Experimentally, the NV ensemble contains NV centers with four orientations. Based on Eq. (3), the broadening of the ODMR spectrum is mainly determined by NV centers with the smallest $\theta$, which have the largest frequency shift induced by the Berry phase(Supplementary Fig. 6). The frequency shift of $\mp\omega_r\cos\theta$ is

insensitive to the angle $\theta$ for small $\theta$. This explains why the theoretical results for $\theta = 0°$ and $\theta = 20°$ are similar, and both agree well with the experimental results. All data shown in Fig. 3b, c are taken from one levitated diamond.

To determine the frequency shift as a function of the rotational frequency unambiguously, an external magnetic field can be applied to separate the energy levels of NV centers along four different orientations. Here we apply a static magnetic field of about 100 G along the $z$-axis to separate energy levels (Fig. 3d). Data shown in Fig. 3e, f are taken from one levitated diamond, which is different from the one used for Fig. 3b, c. In Fig. 3e, the red squares show the ODMR spectrum measured at a rotation frequency of 0.1 MHz. The linewidths of ODMR dips for levitated diamond NV centers are broader than those for fixed NV centers due to the continuous change of NV orientations relative to the magnetic field. Compared with the ODMR spectrum of a levitated nanodiamond without stable rotation (gray circles), the linewidth of each dip for a diamond rotating at 0.1 MHz is narrower. This clearly demonstrates that fast rotation can stabilize the orientation of the levitated nanodiamond. Now we consider the NV centers with the smallest $\theta$ (largest Zeeman shift) and the transition between the state $|m_s = 0\rangle$ and the state $|m_s = +1\rangle$ as an example. The electron spin resonance frequency is 3.120 GHz at $\omega_r = 2\pi \times 0.1$ MHz for this transition. The corresponding angle between the NV-axis and the rotation axis is $\theta = 20.7°$, which is calculated based on the transition frequency and the magnitude of the external magnetic field. We then measure the resonance frequency at different clockwise (unless otherwise specified, all are viewed from the positive $z$ direction) rotation frequencies, as shown in Fig. 3f. The resonance frequency increases following the increase of the rotation frequency. The experimental data points are in between the theoretically calculated curves for $\theta = 20.7°$ (green solid line) and $\theta = 24.0°$ (violet dashed line), indicating the orientation of the NV axes changes slightly when the rotation frequency increases. This is because the electric dipole moment of the levitated nanodiamond is not exactly perpendicular to the axis of the largest or the smallest moment of inertia. Once the rotation frequency increases, the nanodiamond tends to rotate along its stable axis and the driving torque is not large enough to keep its former orientation. The magenta dashed curve is a linear fitting of the resonance frequency. The orientation of the NV center can be calculated by the resonance frequency at the various rotation frequencies. The angle $\theta$ changes by approximately $3.3°$ at $\omega_r = 2\pi \times 10$ MHz, compared with that at $\omega_r = 2\pi \times 0.1$ MHz. A rotating diamond can also serve as a gyroscope[58,59].

The effect of the Berry phase in a levitated nanodiamond rotating at the counterclockwise direction is shown in Supplementary Fig. 6. The resonance frequency between the state $|m_s = 0\rangle$ and the state $|m_s = +1\rangle$ decreases as the rotation frequency increases for counterclockwise rotation (Supplementary Fig. 6c), which is different from that of the levitated nanodiamond rotate clockwise (Fig. 3f).

## Quantum control of fast-rotating NV centers

Quantum control of spins is important for creating superposition states[25,26,39] and performing advanced quantum sensing protocols[60]. Here, we apply a resonant microwave pulse to demonstrate quantum state control of fast-rotating NV centers. The spin state can be read out by measuring the emission PL. Because a weak 532 nm laser is used to avoid significant heating, the initialization time should be long enough to prepare the NV spins to the $|m_s = 0\rangle$ state. When the laser intensity is 0.113 W/mm², the initialization time is 1.05 ms (Supplementary Fig. 7a). This is shorter than the spin relaxation time ($T_1 \sim 3.6$ ms) of this levitated nanodiamond (Supplementary Fig. 7b). We also measure Rabi oscillation of a nanodiamond fixed on a glass cover slip with the same 532 nm laser intensity for comparison. We get similar results for both high and low intensities of the 532 nm laser (Supplementary Fig. 8).

Due to the $\Omega$-shape of the microwave antenna, the orientation of the magnetic field of the microwave is located in the $yz$-plane and slightly different from the $z$-axis with an angle of about $\theta' = 8.5°$ (Fig. 4a). So, $\mathbf{n}_{MW} = (-\sin\theta', 0, \cos\theta')$. The effective microwave magnetic field acting on NV spins, with the orientation of $\mathbf{n}_{NV} = (\cos\phi(t)\sin\theta, \sin\phi(t)\sin\theta, \cos\theta)$, changes as a function of the rotation phase $\phi(t)$ of the levitated nanodiamond. The Rabi frequency $\Omega_{Rabi}$ can be written as[61,62]:

$$\Omega_{Rabi} \propto \sqrt{1 - (\cos\theta\cos\theta' - \sin\phi(t)\sin\theta\sin\theta')^2}. \quad (4)$$

Therefore, it is necessary to synchronize the microwave pulse and the rotation phase of the levitated nanodiamond. Figure 4b shows the pulse sequence of the Rabi oscillation measurement. The time gap between the initialization and the readout laser pulses is twice of the rotation period, which allows us to apply the microwave pulse at an arbitrary rotation phase between 0 and $2\pi$.

The measured Rabi oscillations between the state $|m_s = 0\rangle$ and the state $|m_s = +1\rangle$ of NV centers are shown in Fig. 4d, e. All these measurements are carried out at a rotation frequency of 100 kHz. The rotation period is 10 $\mu$s which is much longer than the microwave pulse. For NV centers with different orientations, the Rabi frequencies are different. The measured Rabi frequencies are 7.10 MHz, 6.57 MHz and 2.80 MHz when the applied microwave frequencies are 2.936 GHz (dip 1), 3.009 GHz (dip 2), and 3.129 GHz (dip 3), respectively (Fig. 4d). Figure 4e shows Rabi oscillations of the NV centers with $\theta = 22°$ at different rotation phases. The blue circles and black squares are measured at the rotation phase of $\phi = \pi/2$ and $\phi = \pi$, respectively. The corresponding Rabi frequencies are 2.72 MHz and 2.23 MHz due to the different projections of the microwave magnetic field along the NV-axis. We also apply microwave pulse at other rotation phases to explore how it affects the Rabi frequency. Figure 4f shows the Rabi frequency $\Omega_{Rabi}$ for NV centers with $\theta = 22°$ (blue circles) as a function of the rotation phase. The Rabi frequency is smallest at $\phi = 3\pi/2$. The red curve is the theoretical prediction, which agrees well with our experimental results.

## Feedback cooling of the center-of-mass motion

To study quantum spin-mechanics and use a levitated diamond for precision measurements, it will be crucial to reduce the energy of the CoM motion of a levitated diamond. Our integrated Paul trap has multiple electrodes, which can be used for feedback cooling. Because of the high quality factor of the CoM in high vacuum and the low frequency of the CoM motion, we add $\pi/2$ phase delays to the position signals of the levitated diamond to obtain its velocity signals. We then apply electric forces on the charged diamond proportional to the velocities but with opposite signs to cool the CoM motion. The schematic diagram is shown in Fig. 5a. The feedback loop is implemented through an FPGA (Field Programmable Gate Array). The motion signals are read out, followed by band-pass filters, amplifiers and phase delayers, and then fed back to the four electrodes at the corners of the ion trap. Figure 5b–d shows the PSDs of the CoM motion of a levitated nanodiamond at the pressure of 0.02 Torr without feedback cooling (noFB, blue curves) and at the pressure of $2.0 \times 10^{-5}$ Torr with feedback cooling (FB, red curves). The orange curves are the corresponding noise floors. Based on the fitting, the final temperature of the CoM motion with feedback cooling are $1.2 \pm 0.3$ K, $3.5 \pm 0.4$ K, and $86 \pm 26$ K along the $x$, $y$, and $z$ directions, respectively. The final temperatures are mainly limited by the small size of the center hole of the surface ion trap used for forward detection (Fig. 5a), which severely limits the NA of the detection system. The cooling efficiency can be improved in the future with backward detection by using the backward scattered light of the levitated diamond collected by the objective lens.

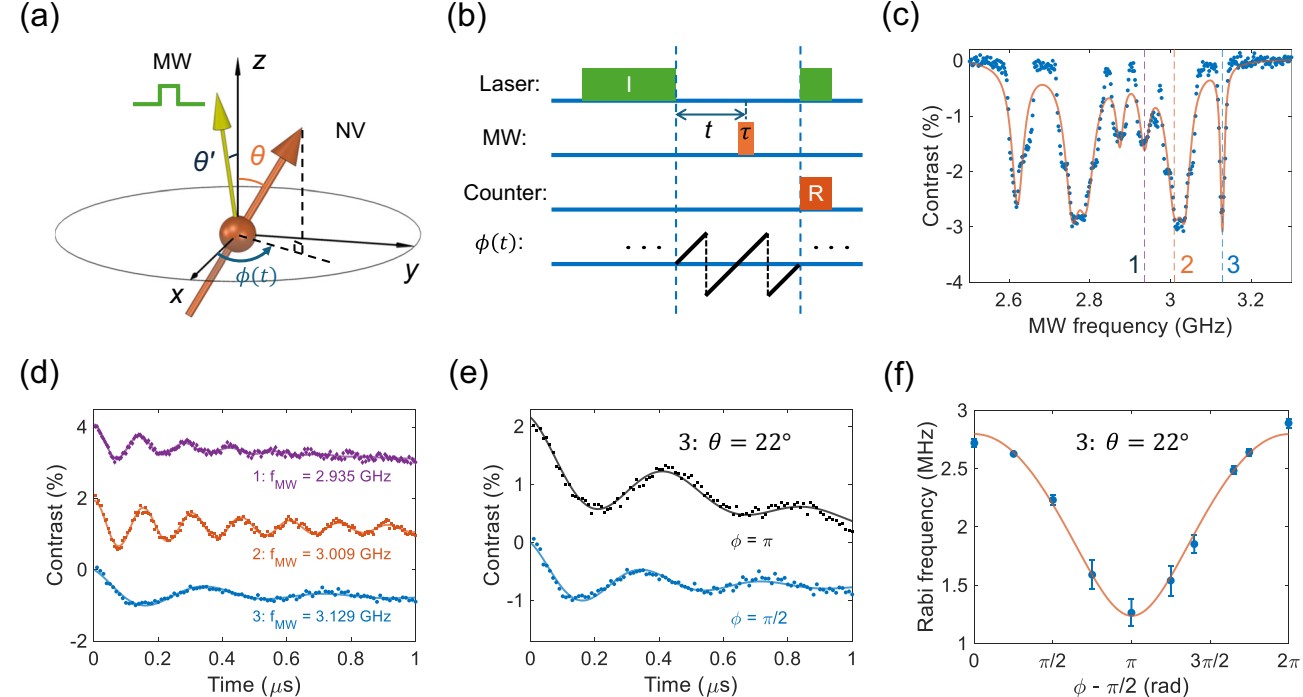

**Fig. 4 | Quantum control of NV centers in a levitated nanodiamond in high vacuum with a rotation frequency of 100 kHz. a** Schematic of the Rabi oscillation measurement at different rotation phase $\phi(t)$. The angle $\theta'$ between the magnetic component of microwave and the $z$-axis is 8.5°. **b** Pulse sequence of the Rabi oscillation measurement. **c** ODMR of the levitated nanodiamond. **d** Measured Rabi oscillations of NV centers at three different orientations. The Rabi frequencies are 7.10 MHz, 6.57 MHz, and 2.80 MHz at the ODMR frequencies of 2.935 GHz, 3.009

GHz, and 3.129 GHz, respectively. The red and purple curves are shifted 2% and 4% to separate the curves. **e** Rabi oscillation of NV centers with $\theta = 22°$ corresponding to the resonance frequency of 3.129 GHz (dip 3). The blue circles and black squares are measured at rotation phase of $\phi = \pi/2$ and $\phi = \pi$, respectively. The black curve is shifted 2%. **f** Rabi frequency at $\theta = 22°$ (blue circles) as a function of the rotation phase $\phi$. The red curve is the theoretical prediction. The error bars represent standard deviations among three measurements.

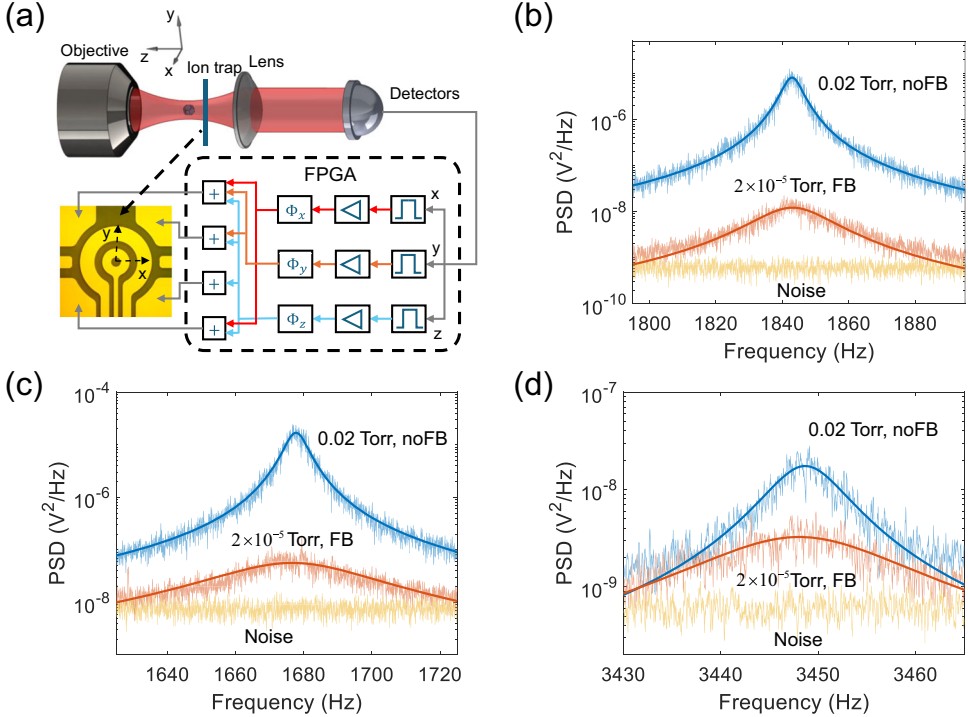

**Fig. 5 | Feedback cooling of the CoM motion of a levitated nanodiamond in the ion trap. a** Schematic diagram of the feedback cooling method. **b**–**d** PSDs of the CoM of the levitated nanodiamond along the (**b**) $x$, (**c**) $y$, and (**d**) $z$ directions at the pressure of 0.02 Torr without cooling (blue curves) and at the pressure of $2.0 \times 10^{-5}$

Torr with feedback cooling (red curves). The orange curves are the noise floors. Based on the fitting, the effective temperature of the CoM motion with feedback cooling are $1.2 \pm 0.3$ K, $3.5 \pm 0.4$ K, and $86 \pm 26$ K along the $x$, $y$, and $z$ directions, respectively.

## Discussion

In conclusion, we have levitated a nanodiamond at pressures below $10^{-5}$ Torr with a surface ion trap. We performed ODMR measurement of a levitated nanodiamond in high vacuum for the first time. The internal temperature of the levitated nanodiamond remains stable at about 350 K when the pressure is below $5 \times 10^{-5}$ Torr, which means stable levitation with an ion trap will not be limited by heating even in ultrahigh vacuum. This offers a unique platform for studying fundamental physics, such as massive quantum superposition[25,26,39].

Additionally, we apply a rotating electric field that exerts a torque on the levitated nanodiamond to drive it to rotate at high speeds up to 20 MHz. 20 MHz rotation can generate a pseudo-magnetic field of 0.71 mT for an electron spin, and a pseudo-magnetic field of 6.5 T for an $^{14}$N nuclear spin. With this method, the rotation frequency of a levitated nanodiamond is extremely stable and easily controllable. The effect of the Berry phase generated by rotation[35] is observed with the embedded NV center electron spins. This will be useful for creating a gyroscope for rotation sensing[36,58,59]. We also demonstrate quantum control of rotating NV centers in high vacuum, which will be important for using spins to create nonclassical states of mechanical motion[25,26,39]. Using feedback cooling, the CoM of the levitated nanodiamond is cooled in all three directions with a minimum temperature of about 1.2 K along one direction.

The maximum rotation frequency in this experiment is limited by the bandwidth of the multichannel waveform generation system for generating the phase-shifted signals on the four electrodes. The rotation frequency can be much higher with a better waveform generation system. Furthermore, in the presence of a DC external magnetic field, the NV centers within a rotating nanodiamond experience an AC magnetic field. Quantum sensing of an AC magnetic field can have a higher sensitivity compared to that of a DC magnetic field[63]. Consequently, the mechanical rotation can enhance the sensitivity of a magnetometer in measuring DC magnetic fields. By using purer diamond particles, i.e. CVD diamonds, a higher excitation power of the 532 nm laser can be employed to reduce the initialization time of NV centers.

## Methods

### Experiment setup and materials

The surface ion trap is fabricated on a sapphire wafer by photolithography. The chip is fixed on a 3D stage and installed in a vacuum chamber. The AC high voltage signal used to levitate nanoparticles and the microwave used for quantum control are combined with a bias tee to be delivered to the chip. A 532 nm laser beam is incident from the bottom to excite diamond NV centers. The photoluminescence (PL) is collected by an objective lens with a numerical aperture (NA) of 0.55. A 1064 nm laser beam focused by the same objective lens is used to monitor both the center-of-mass (CoM) motion and the rotation of the levitated nanoparticle. The PL is separated with the 532 nm laser and the 1064 nm laser by dichroic mirrors. The counting rate and optical spectrum of the PL are measured by a single photon couting module and a spectrometer. The processes of particle launching and trapping are monitored by two cameras.

The diamond particles were acquired from Adamas Nano. The product model is MDNV1umHi10mg (1 micron Carboxylated Red Fluorescence, 1 mg/mL in DI Water, ~3.5 ppm NV). The experimental data shown in the main text of the manuscript are obtained from four different diamond particles. The data presented in Fig. 1, Fig. 2, and Fig. 3a–c originate from measurements conducted on the same nanodiamond particle. Figure 3d–f shows the data from a second nanodiamond particle, while the data in Fig. 4 is measured using the third nanodiamond particle. Figure 5 uses the fourth diamond particle.

### Internal temperature of a levitated nanodiamond

In the experiment, we measure the ODMR of levitated nanodiamond NV centers to detect the internal temperature in the absence of an external magnetic field. The zero-field Hamiltonian of NV center is: $H = DS_z^2/\hbar + E\left(S_x^2 - S_y^2\right)/\hbar$, where $D$ is the zero-field energy splitting between the states of $|m_s = 0\rangle$ and $|m_s = \pm 1\rangle$, $E$ is the splitting between the states due to the strain effect. The small splitting between two dips in the ODMR spectra (Fig. 1e) without an external magnetic field is due to the $E$ term from strain in the nanodiamond. The zero-field splitting $D$ is dependent on temperature[41,50]:

$$D = c_0 + c_1 T + c_2 T^2 + c_3 T^3 + \Delta_{pressure} + \Delta_{strain}, \qquad (5)$$

where $c_0 = 2.8697$ GHz, $c_1 = 9.7 \times 10^{-5}$ GHz/K, $c_2 = -3.7 \times 10^{-7}$ GHz/K$^2$, $c_3 = 1.7 \times 10^{-10}$ GHz/K$^3$, $\Delta_{pressure} = 1.5 \times 10^{-6}$ GHz/bar, and $\Delta_{strain}$ is caused by the internal strain effect. $\Delta_{pressure}$ is smaller and can be neglected in vacuum. Figure 1e is the ODMR measured at the pressure of 10 Torr (blue circles) and $6.9 \times 10^{-6}$ Torr (red squares). The zero-field splitting obtained by fitting can be used to calculate the temperature of the levitated nanodiamond.

The internal temperature $T$ of a levitated nanodiamond is determined by the balance between heating and cooling effects[51,52]:

$$A_a = A_{gas}p(T - T_0) + A_{bb}\left(T^5 - T_0^5\right), \qquad (6)$$

where $A_a = \sum_\lambda \eta_\lambda I_\lambda V$ is the heating of the excitation laser ($\lambda = 532$ nm) and the detecting laser ($\lambda = 1064$ nm), $\eta_\lambda$ is the absorption coefficient of nanodiamond and $I_\lambda$ is the laser intensity, $V$ is the volume of nanodiamond. The first term at the right side of the equation is the cooling rate caused by gas molecule collisions, $A_{gas} = \frac{1}{2}\kappa\pi R^2 \upsilon T_0 \frac{\gamma'+1}{\gamma'-1}$, $\kappa \approx 1$ is the thermal accommodation coefficient, $R$ is the radius of nanodiamond, $\upsilon$ is the mean thermal speed of gas molecules, $\gamma'$ is the specific heat ratio ($\gamma' = 7/5$ for air near room temperature), $p$ is the pressure, $T_0$ is the thermal temperature. The last term is the cooling rate of black-body radiation. $A_{bb} = 72\zeta(5)Vk_B^5/\left(\pi^2 c^3 \hbar^4\right)\mathrm{Im}\left(\frac{\varepsilon-1}{\varepsilon+2}\right)$, where $\zeta(5) \approx 1.04$ is the Riemann zeta function, $k_B$ is the Boltzmann constant, $c$ is the vacuum light speed, $\hbar$ is the reduced Planck's constant, $\varepsilon$ is a constant and time-independent permittivity of nanodiamond across the black-body radiation spectrum. By measuring the internal temperature as a function of the intensities of the 532 nm laser and the 1064 nm laser, the absorption coefficients of the nanodiamond are estimated to be 111 cm$^{-1}$ at 532 nm and 5.87 cm$^{-1}$ at 1064 nm (Supplementary Fig. 3).

### Reporting summary

Further information on research design is available in the Nature Portfolio Reporting Summary linked to this article.

## Data availability

Source data for figures in the main text are provided with this paper in the Source Data file. Other data that support the findings of this study are available from the corresponding author upon request. Source data are provided with this paper.

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

## Acknowledgements

The authors thank Jun Ye for helpful discussions. T.L. acknowledges the support from the National Science Foundation under Grant PHY-2110591, the Office of Naval Research under Grant No. N00014-18-1-2371, and the Gordon and Betty Moore Foundation, grant DOI 10.37807/gbmf12259. This project is also partially supported by the Laboratory Directed Research and Development program at Sandia National Laboratories, a multimission laboratory managed and operated by National Technology and Engineering Solutions of Sandia LLC, a wholly owned subsidiary of Honeywell International Inc., for the U.S. Department of Energy's National Nuclear Security Administration under Contract No. DE-NA0003525. This paper describes objective technical results and analysis. Any subjective views or opinions that might be expressed in the paper do not necessarily represent the views of the U.S. Department of Energy or the United States Government. C.Z. acknowledges the support by NSF ExpandQISE 2328837.

## Author contributions

T.L., Y.J., K.S., and P.J. conceived and designed the project. Y.J. and K.S. built the setup. Y.J. performed measurements and calculations. Y.J., K.S., T.L., X.G., C.Z., and A.J.G. discussed the results. T.L. supervised the project. All authors contributed to the writing of the manuscript.

## Competing interests

The authors declare no competing interests.
