## [Peer Review File · Nature Communications]

REVIEWER COMMENTS

Reviewer #1 (Remarks to the Author):

Jin et al report on electrical trapping of nano-diamonds containing NV centres in high vacuum. Because the vacuum pressure is so low, they are able to physically rotate the nano-diamonds at up to 20MHz, and demonstrate measurement of "fictitious magnetic fields" and NV spin manipulations. They show that the nano-diamonds heat up, but do not leave the trap. This paper reports a significant experimental advance and is arguably a groundbreaking moment for the study of rotating quantum systems and levitodynamics, enabling a swath of new fundamental experiments. However, the actual paper is in need of significant revision before publication, as we have identified several important issues that must be addressed. The citation to previous work in the field is also in need of improvement.

1. The most pressing issue is the author's identification of the "magnetic pseudo field", in both terminology and in their experimental results. Regarding terminology, there are important differences between the "Barnett field", the "magnetic pseudo-field" and what the authors measure in this paper. The Barnett effect is magnetisation of spins, usually electron spins, by rotation, and while the vector quantity $B = \omega/\gamma$ defines an effective magnetic field that defines this magnetisation, the measured magnetic field that results *from the rotationally-induced magnetisation* is sometimes called the "Barnett field". This field is tiny even for very high rotation speeds, since rotation at ω induces a pseudo field B that then induces a polarisation $M = \mu \exp(-\hbar \omega / k_B T)$. For this reason, the term "magnetic pseudo field" is often preferred, since one could mistake measurement of a "Barnett field" with measurement of the "Barnett effect". Atomic spin gyroscopes measure such a fictitious magnetic field and do not call it a Barnett field.

2) The "magnetic pseudo field" cannot be measured unless the measurement, in this case the microwave fields being applied, are in the rotating frame as well. This is the principle of magnetic spin resonance gyroscopes, and the situation is mathematically equivalent to that of a Foucault pendulum: an observer on the rotating earth sees the pendulum precess (at the north pole), while a hypothetical observer in space, in a fixed absolute frame of reference would not see any rotation of the pendulum's precession plane. In rotating NMR/NQR experiments, the precessing spins and sensitivity to different angular momentum observables result in a "rotational Doppler shift" (Phys. Rev. B 103, 174308 2021) that isn't relevant here since the NV transduces the interaction with the microwave field into a change in PL, which is frame independent. The authors are suggested to consult Ref. 27 more closely, where the pseudo field applied to C13 spins is detected by the NV centres (both in rotating frame) and Chudo et al 2014 Appl. Phys. Express 7 063004 where the pseudo field (called the Barnett field in this paper) is detected by NMR with a rotating NMR detection coil. Recent work by the same team (Phys. Rev. B 103, 174308 2021) goes some way to clarifying the differences between the various fictitious fields, including the "rotational Doppler effect" apparent in

rotating NMR/NQR measurements. The authors should also consult and cite R. Tycko Phys. Rev. Lett. 58, 2281 1987 where essentially the same experiment is done.

3) Equations 1, 2, and 3 are incorrect, or at the very least it is improperly stated what frame they are in. If this is the NV hamiltonian "in the rotating frame", they are missing the terms that arise when the unitaries ($R(\phi)$) are time dependent, i.e a term $i\hbar d R(t)/dt R^*(t)$ is missing from each. Evaluation of this term yields the magnetic pseudo field term, $\hbar\omega S_z$, which consequently *only appears in the rotating frame*. Furthermore, Eq 1 (and 2 and 3) are taken from Ref 32, where several additional transformations are made to the Hamiltonian, including one where rotation of the coordinate axes are eliminated, these are not mentioned at all, other than "...eq 2 is obtained via a unitary transform". Eq 2 does not trivially follow from Eq 1, and in any case the equivalent to Eq. 1 in Ref 32 is also different to that in the current paper, though there is an error in Ref 32 describing these transformations. Equation 3 is also somewhat dubious: the transformation as written implies that the rotation operators are applied to the lab-frame magnetic field, which they are not. The authors need to take care which frame they are working in and expressing their equations in. The simplest way to derive Eq 2 (and correct Eq 3) is to consider the NV in it's natural, diagonal basis and transform the magnetic field into the rotating frame, $H' = R(t)^{-1} \cdot (B \cdot S) \cdot R(t) + i\hbar d R(t)^{-1} / dt \cdot R(t)$. Note the different order of the transformation since we are going from the lab frame to the rotating frame. With $B = 0$ one recovers Eq 2 in the manuscript. One should also make careful note that this does not include the effect of a microwave field which ultimately measures the NV frequency shift.

4) However, the authors do measure a frequency shift that arises from rotation, and attribute it to the magnetic pseudo field, or rather the vector projection of the pseudo field. Actually this is from Berry's phase. The key point here is that Berry's phase breaks the symmetry between the stationary and rotating frames, and is able to be detected in both. The Berry phase will induce a frequency shift of $\omega(1 - \cos(\theta))$, proportional to the solid angle swept out by the NV axis (see eg Eq 1 in Ref 33), and if $\theta = 0$ then there will be no frequency shift. Indeed, the Berry phase and the pseudo field are related by a constant ω picked up moving from one frame to another, hence the $\omega - \omega \cos(\theta)$ form. I think this change may result in the identified angles changing.

5) The authors describe the measurement in fig 2e as detecting the 'pseudo field' via a broadening of the ODMR features, while in Fig 3b they add a real magnetic field that splits the NV orientation classes so they can then resolve individual transitions and hence make a measurement of the frequency shifts imparted by rotation. This is confusing, since the measurements are essentially identical except with the addition of a bias magnetic field, and the paper reads as if the frequency shifts accumulated in the zero field case are somehow different to that when an applied field is considered (i.e. broadening vs splitting), whereas it's exactly the same and in the former they can't resolve individual transitions. I do not see the point of the zero field data when they then do the applied field experiment, which is much cleaner and more easily interpreted. These could be combined into the same figure, or the zero-field data omitted (and Eqs 1, 2, 3 revised). The significance of Eq 2 isn't well argued either, since the raising-lowering terms (the transverse magnetic

field terms) can generally be ignored, they are too small even here to result in significant NV spin state mixing.

6) The analysis for the data presented in Fig 2(f) and Fig 3(c) seems very simple to the point of trivial and thus not very informative at all. For Fig 2f, why are two (random) angles (0, 20deg) chosen, when the 4 orientation classes of NVs in the diamond would each make a separate angle to the rotation axis? The authors say in the manuscript that the broadening is determined mainly by NVs closer to the rotation axis, but I can't see why the authors couldn't use the inferred orientations deduced from the magnetic field data later? For Fig 3e A quantity of interest such as the rotation frequency when the angle changes could be deduced without much effort. Perhaps a piecewise linear fit?

7) Key details regarding the diamond sample and experiment are entirely omitted, and these undermine the claims of the paper significantly. For example while the authors mention T1 as being 3.6ms, this is not the time that matters for coherent quantum control: what is T2, T2*? The supplement describes a measurement of "T2_{rabi}", which is not a very meaningful metric of the spin coherence of the NVs (it says just as much about the stability of the mw field as the inhomogeneous broadening of the NVs). The authors claim that this work spins diamonds "about three orders of magnitudes faster than prior achievements using diamonds mounted on motor spindles ... this rotation speed surpasses the dephasing rate of NV spins in the diamond" but that has not been shown until we know the T2, T2* of the NVs in this diamond. In particular, the demonstration of Rabi oscillations was only possible when the rotation speed was reduced to 100kHz, so T2, T2* would need to be 10 μ s, which is rarely the case in nano-diamonds. Additionally, it is unclear if all the measurements were taken on one nano-diamond (impressive if the case) or if many nano-diamonds were sacrificed in obtaining these results. This is important in the context of reproducibility. For example the change in orientation angle as a function of rotation I would imagine would be different for each nano-diamond.

8) The angular variation of the Rabi frequency due to the angle of the microwave field to the rotation axis is a well-characterised effect that does not change with rotation speed, the authors should consult and cite A. A. Wood et al Physical review letters 124 (2), 020401 2020, where the angular variation of the effective microwave phase is also discussed in detail, as well as in A. A. Wood et al Physical Review Research 3 (4), 043174 2021 where angular variation of the microwave and rf rabi frequency is characterised and corrected for.

9) The discussion and outlook of the work is quite disappointing. There is no discussion of the limitations in the experiment that need to be overcome or subject to further work, eg. the consequences of Rabi frequencies on par with the rotation speed, the tradeoff between green laser intensity and NV spin preparation time (1ms at 100kHz says there's a lot more work to do). Are higher rotation speeds possible? What are the effects of the rotation on the ¹⁴N hyperfine interaction, or ¹³C spins in the diamond (which at 20MHz see a magnetic pseudo field of almost 2T

and should start to polarise, creating a real Barnett field!). The authors might also like to consider how their results impact the other applications rapidly rotating NV centres enables, eg. improved magnetometry A. A Wood et al Physical Review Applied 18 (5), 054019 2022. Regarding NV based gyro sensors, the actual demonstrated work of Soshenko et al Phys. Rev. Lett. 126, 197702 2021 and Jarmola et al Sci. Adv 7 eabl3840 2021 should be cited.

10) Finally, I would say there is a sense of sloppiness in the preparation of the manuscript, with many typos and errors (discussed above). The many typos are perhaps best exemplified by "Date Availability" instead of "Data Availability". This is followed by what appears to me to be a meaningless statement "All data generated and analyzed in this study are available within the article and its Supplementary Information." Does this mean the data is not available upon request?

To summarise this manuscript reports a significant experimental advance and is arguably a groundbreaking moment for the study of rotating quantum systems and levitodynamics, enabling a swath of new fundamental experiments. However, I would only recommend publication after ALL of the above 10 points have been fully addressed

Reviewer #2 (Remarks to the Author):

Jin et al levitate nanodiamonds using a surface ion trap and take nanodiamonds to high vacuum. They then demonstrate nitrogen-vacancy ODMR, fast rotation of the nanodiamond, and measure the spin coherence time. This is indeed the first demonstration of ODMR at high vacuum which has been an important obstacle. This is of wide interest to the specific growing community of levitated nanoparticles as well as the NV community. In addition, it will be of interest to the wider quantum technology community. The work supports its conclusions and claims. The methodology is sound. Overall, it would make a good paper for Nature Communications if the following questions can be answered satisfactorily:

Jin et al levitate nanodiamonds using a surface ion trap and take them to high vacuum. They then demonstrate nitrogen-vacancy ODMR including Rabi oscillations and fast rotation of the nanodiamond. This is indeed the first demonstration of ODMR of a levitating particle at high vacuum which has been an important obstacle. The extra things are excellent also: measurements where the diamond is spinning faster than the NV spin decoherence is a new regime: 1000 times faster spinning than previous NV ODMR. The Rabi oscillations are the first quantum control of spins while levitated in high vacuum. This work is of great importance to the specific fast-growing community of levitated nanoparticles, which is working on applications in future quantum sensors and tests of fundamental physics, such as for studying the quantum nature of gravity. The NV community will want to know about this work also. In addition, it will be of interest to the wider quantum technology community.

The experimental work supports the conclusions and claims. The methodology is sound. Overall, it would make a great paper for Nature Communications if the following questions can be answered satisfactorily:

1. The NV ODMR linewidth (~ 30 MHz) of the levitated diamonds in figure 1e and 2e is large compared to the established literature and compared to the measurements with the nanodiamonds on a substrate (supplementary info figure 8a). In figure 4c the ODMR looks sharper and more like a simple NV ODMR spectrum. Some explanation of this should be given in the manuscript.
2. The manuscript should describe the type of nanodiamond used e.g. is it purchased, isotopically purified, HPHT or CVD grown etc.
3. It would be interesting to cool the centre-of-mass (CM) motion of the nanodiamonds. However, the authors do not do so despite the fact that they have significant expertise in this field. Is there some reason for this?
4. The lowest pressure the authors have achieved is 6.9×10^{-6} Torr. Is there any limitation here for why the authors didn't go further down in pressure?

Some further comments to improve the manuscript:

- A. It should be mentioned that the Stark shift for NV would be small and so is not relevant here.
- B. In Figure 1e there is a splitting due to the E term coming from strain in the nanodiamonds. This point should be explained for non-experts.
- C. In Figure 2d the caption should say that the nanodiamond is rotating about the z axis, not along the z axis.

ATM Anishur Rahman and Gavin W Morley

Reviewer #3 (Remarks to the Author):

I co-reviewed this manuscript with one of the reviewers who provided the listed reports. This is part of the Nature Communications initiative to facilitate training in peer review and to provide appropriate recognition for Early Career Researchers who co-review manuscripts

Reviewer #5 (Remarks to the Author):

Recommendation: This paper is publishable subject to minor revisions noted. Further review is optional.

Comments:

This paper presents exciting results on the quantum control and fast rotation of levitated nanodiamonds (NDs) containing NV centers in high vacuum. The authors developed a planar Paul trap with an integrated microwave antenna to trap nanodiamonds down to 10^{-5} Torr. They show that they can rotate the NDs up to 20MHz, faster than the dephasing rate of the NVs. They take advantage of the NVs to estimate the particle's internal temperature and probe the pseudo magnetic field generated by the rotation, the Barnett effect. Finally, they show coherent control of the NVs while the ND is rotated at 0.1MHz. This work sets a new milestone for the levitated optomechanics community. It shows a promising technical solution for applied and fundamental physics such as gyroscope, the generation of massive quantum superposition using spins or the detection of quantum gravity. I recommend this article for publication in Nature Communication, provided that the authors address my questions.

(1) In the abstract, the authors claim that “fast rotation of levitated diamond has not been reported.” This statement should be removed as it is not quantitative. What is considered fast here?

(2) Introduction, paragraph 2, regarding the levitation of diamonds at low vacuum level, the author should add the following reference: M. C. O'Brien & al., Appl. Phys. Lett. 114, 053103 (2019)

(3) Introduction, paragraph 3, line 3, what did the authors mean by “it compromises multiple electrodes...”?

(4) Introduction, paragraph 3, second to last phrase. I suggest changing the phrase “we achieve the quantum coherent control ...”. Is the control coherent or quantum? Can it be quantum without being coherent and vice-versa?

(5) In Figure 1, I need help finding the method used for the electric field simulation in the supplementary. Add a phrase or two on the method used, at least in the supplementary.

(6) Part C, first phrase: “Quantum coherent control...”, same remark as in (4), I would suggest using either quantum or coherent.

(7) Fig 4. Title, same remark as above.

(8) In Figure 4e, I suggest using a different color scheme than the one in Fig. 4d to avoid confusing readers. Which transitions (1, 2 or 3) correspond to $\theta=22$ degree? Make it clear in the legend and caption.

(9) Part C, formula (4) It would make the article and statement more straightforward if the authors added a small discussion (one phrase or two) on why the Rabi frequency is affected by the drive

phase ϕ . The authors mentioned it in the supplementary, but moving it to the main text will be better.

(10) The authors should add a reference to the following article as it is very much related to the authors' article: M. Perdriat & al., "Spin Read-out of the Motion of Levitated Electrically Rotated Diamonds", arXiv:2309.01545 (2023)

Reply to Reviewer #1:

Jin et al report on electrical trapping of nano-diamonds containing NV centres in high vacuum. Because the vacuum pressure is so low, they are able to physically rotate the nano-diamonds at up to 20MHz, and demonstrate measurement of "fictitious magnetic fields" and NV spin manipulations. They show that the nano-diamonds heat up, but do not leave the trap. This paper reports a significant experimental advance and is arguably a groundbreaking moment for the study of rotating quantum systems and levitodynamics, enabling a swath of new fundamental experiments. However, the actual paper is in need of significant revision before publication, as we have identified several important issues that must be addressed. The citation to previous work in the field is also in need of improvement.

Reply: We thank the Reviewer for the positive assessment of the significance of our work and helpful comments. We have revised the manuscript according to the Reviewer's comments and suggestions. Point-by-point responses to the Reviewer's comments and questions are provided below.

Comment 1) The most pressing issue is the author's identification of the "magnetic pseudo field", in both terminology and in their experimental results. Regarding terminology, there are important differences between the "Barnett field", the "magnetic pseudo-field" and what the authors measure in this paper. The Barnett effect is magnetisation of spins, usually electron spins, by rotation, and while the vector quantity $B = \omega/\gamma$ defines an effective magnetic field that defines this magnetisation, the measured magnetic field that results *from the rotationally-induced magnetisation* is sometimes called the "Barnett field". This field is tiny even for very high rotation speeds, since rotation at ω induces a pseudo field B that then induces a polarisation $M = \mu N \exp(-\hbar \omega / k_B T)$. For this reason, the term "magnetic pseudo field" is often preferred, since one could mistake measurement of a "Barnett field" with measurement of the "Barnett effect". Atomic spin gyroscopes measure such a fictitious magnetic field and do not call it a Barnett field.

Reply: We appreciate the valuable comments provided by the Referee. Following comments 1-4 of the referee, we have revised all descriptions related to the observed frequency shift and attributed it to the Berry phase in the laboratory frame. In addition, we modified the title of our manuscript to "*Quantum control and Berry phase of electron spins in rotating levitated diamonds in high vacuum.*" We also added a paragraph to clarify the relation between the pseudo-magnetic field and the Berry phase:

"The fast-rotating diamond with embedded NV spins allows us to observe the effects of the Berry phase due to mechanical rotation. The Berry phase, also known as the geometric phase, is a fundamental aspect of quantum mechanics with applications in multiple fields, including the

topological phase of matter and the quantum hall effect [53–57]. The Berry phase in the laboratory frame is equivalent to the pseudo-magnetic field (called the Barnett field in [57]) in the rotating frame: $B_\omega = \omega_r/\gamma$, where γ is the spin gyromagnetic ratio. In this work, the microwave source is fixed in the laboratory frame. Only the levitated diamond is rotating. So, we can observe the effect of the Berry phase due to rotation [57].”

Comment 2) The "magnetic pseudo field" cannot be measured unless the measurement, in this case the microwave fields being applied, are in the rotating frame as well. This is the principle of magnetic spin resonance gyroscopes, and the situation is mathematically equivalent to that of a Foucault pendulum: an observer on the rotating earth sees the pendulum precess (at the north pole), while a hypothetical observer in space, in a fixed absolute frame of reference would not see any rotation of the pendulum's precession plane. In rotating NMR/NQR experiments, the precessing spins and sensitivity to different angular momentum observables result in a "rotational Doppler shift" (Phys. Rev. B 103, 174308 2021) that isn't relevant here since the NV transduces the interaction with the microwave field into a change in PL, which is frame independent. The authors are suggested to consult Ref. 27 more closely, where the pseudo field applied to C13 spins is detected by the NV centres (both in rotating frame) and Chudo et al 2014 Appl. Phys. Express 7 063004 where the pseudo field (called the Barnett field in this paper) is detected by NMR with a rotating NMR detection coil. Recent work by the same team (Phys. Rev. B 103, 174308 2021) goes some way to clarifying the differences between the various fictitious fields, including the "rotational Doppler effect" apparent in rotating NMR/NQR measurements. The authors should also consult and cite R. Tycko Phys. Rev. Lett. 58, 2281 1987 where essentially the same experiment is done.

Reply: (i) As mentioned in our reply to Comment 1, we have revised all descriptions related to the observed frequency shift and attributed it to the Berry phase in the laboratory frame instead of the pseudo-magnetic field.

(ii) We have read the mentioned references, especially [Phys. Rev. B 103, 174308 (2021)], carefully. The references considered three situations: (a) Only the coil is rotating; (b) Both the sample and the coil undergo rotation; (c) Only the sample is rotating. In our experiment, only the levitated diamond is rotating, and the stationary microwave field is essentially parallel to the rotation axis. Thus, our setup aligns with the third case (c). Consequently, the observed frequency shift is attributed to the Berry phase in the laboratory frame.

(iii) We have cited the mentioned references: [57] Phys. Rev. B 103, 174308 (2021); [31] Appl. Phys. Express 7, 063004 (2014); and [53] Phys. Rev. Lett. 58, 2281 (1987).

Comment 3) Equations 1, 2, and 3 are incorrect, or at the very least it is improperly stated what frame they are in. If this is the NV hamiltonian "in the rotating frame", they are missing the terms that arise when the unitaries ($R(\phi)$) are time dependent, i.e a term $i\hbar d R(t)/dt R^*(t)$ is missing

from each. Evaluation of this term yields the magnetic pseudo field term, $\hbar\omega S_z$, which consequently *only appears in the rotating frame*. Furthermore, Eq 1 (and 2 and 3) are taken from Ref 32, where several additional transformations are made to the Hamiltonian, including one where rotation of the coordinate axes are eliminated, these are not mentioned at all, other than "...eq 2 is obtained via a unitary transform". Eq 2 does not trivially follow from Eq 1, and in any case the equivalent to Eq. 1 in Ref 32 is also different to that in the current paper, though there is an error in Ref 32 describing these transformations. Equation 3 is also somewhat dubious: the transformation as written implies that the rotation operators are applied to the lab-frame magnetic field, which they are not. The authors need to take care which frame they are working in and expressing their equations in. The simplest way to derive Eq 2 (and correct Eq 3) is to consider the NV in its natural, diagonal basis and transform the magnetic field into the rotating frame, $H' = R(t)^{-1} \cdot (B \cdot S) \cdot R(t) + i \hbar \frac{d}{dt} R(t)^{-1} \cdot R(t)$. Note the different order of the transformation since we are going from the lab frame to the rotating frame. With $B = 0$ one recovers Eq 2 in the manuscript. One should also make careful note that this does not include the effect of a microwave field which ultimately measures the NV frequency shift.

Reply: (i) We have revised equations 1-3. We have also included a step-by-step calculation of the Hamiltonian in Section IV of the Supplementary Information. The essential results derived from this calculation are presented in the main text.

(ii) The new Equation (1) shows the Hamiltonian of the rotating NV electron spin in the laboratory frame:

$$H_{lab} = H_{0,lab} + g\mu_B B S_z = \frac{1}{\hbar} R(t) D S_z^2 R^\dagger(t) + g\mu_B B S_z$$

$$= D \hbar \begin{pmatrix} \cos^2 \theta + \frac{\sin^2 \theta}{2} + \frac{g\mu_B B}{D} & \frac{e^{-i\phi} \cos \theta \sin \theta}{\sqrt{2}} & \frac{e^{-2i\phi} \sin^2 \theta}{2} \\ \frac{e^{i\phi} \cos \theta \sin \theta}{\sqrt{2}} & \sin^2 \theta & -\frac{e^{-i\phi} \cos \theta \sin \theta}{\sqrt{2}} \\ \frac{e^{2i\phi} \sin^2 \theta}{2} & -\frac{e^{i\phi} \cos \theta \sin \theta}{\sqrt{2}} & \cos^2 \theta + \frac{\sin^2 \theta}{2} - \frac{g\mu_B B}{D} \end{pmatrix}.$$

The Hamiltonian possesses three eigenstates $|m_s, t\rangle_{lab}$ ($m_s = 0, \pm 1$). Based on the description of the Berry phase of the eigenstates of a rotating NV center (rotating around the z axis with a constant θ), the Berry phase can be calculated as (Equation 2):

$$\gamma_{m_s} = i \int_0^t \langle m_s, t' | \frac{\partial}{\partial t'} | m_s, t' \rangle_{lab} dt' = m_s \omega_r t \cos \theta$$

Here the Berry phase is calculated for an open-path and is hence gauge-dependent. The spin state of the NV center is observed through the interaction with a microwave magnetic field. In our experiment, the direction of the microwave is in the yz-plane and has a slight angle $\theta' = 8.5^\circ$ relative to z axis, resulting from the asymmetric design of the waveguide. The dominant transition probability arises from the longitudinal (z) component. The expected value of the spin states interacting with the microwave can be expressed as (Equation 3):

$$\begin{aligned} & {}_{lab}\langle \pm 1, t | e^{iH_{lab}t/\hbar} e^{-i\gamma_{\pm 1}} H_{MW,z,lab} e^{i\gamma_0} e^{-iH_{lab}t/\hbar} | 0, t \rangle_{lab} \\ &= \frac{1}{2} g \mu_B B_{MW} \cos \theta' e^{i(-\omega_{MW} + D \pm g \mu_B B \cos \theta \mp \omega_r \cos \theta)t} {}_{lab}\langle \pm 1, 0 | e^{i\theta S_y} S_z e^{-i\theta S_y} | 0, 0 \rangle_{lab} \end{aligned}$$

According to this equation, the transition of the spin state from $|m = 0\rangle$ to $|m = \pm 1\rangle$ can be driven by a microwave operated at the resonance frequency of $D \pm g \mu_B B \cos \theta \mp \omega_r \cos \theta$, where the frequency shift $\mp \omega_r \cos \theta$ is attributed to the Berry phase induced by the mechanical rotation.

Comment 4) However, the authors do measure a frequency shift that arises from rotation, and attribute it to the magnetic pseudo field, or rather the vector projection of the pseudo field. Actually this is from Berry's phase. The key point here is that Berry's phase breaks the symmetry between the stationary and rotating frames, and is able to be detected in both. The Berry phase will induce a frequency shift of $\omega (1 - \cos(\theta))$, proportional to the solid angle swept out by the NV axis (see eg Eq 1 in Ref 33), and if $\theta = 0$ then there will be no frequency shift. Indeed, the Berry phase and the pseudo field are related by a constant ω picked up moving from one frame to another, hence the $\omega - \omega \cos(\theta)$ form. I think this change may result in the identified angles changing.

Reply: (i) We agree with the Referee's comments. We have revised related descriptions and attribute the observed frequency shift to the Berry phase.

(ii) We have included a detailed calculation of the Hamiltonian in the revised version of the Supplementary Information. The transitions induced by the longitudinal (z) and transverse (y) components of the microwave field are:

$$\begin{aligned} & {}_{lab}\langle \pm 1, t | e^{iH_{lab}t/\hbar} e^{-i\gamma_{\pm 1}} H_{MW,z,lab} e^{i\gamma_0} e^{-iH_{lab}t/\hbar} | 0, t \rangle_{lab} \\ &= \frac{1}{2} g \mu_B B_{MW} \cos \theta' e^{i(-\omega_{MW} + D \pm g \mu_B B \cos \theta \mp \omega_r \cos \theta)t} {}_{lab}\langle \pm 1, 0 | e^{i\theta S_y} S_z e^{-i\theta S_y} | 0, 0 \rangle_{lab} ; \end{aligned}$$

$$\begin{aligned} & {}_{lab}\langle \pm 1, t | e^{iH_{lab}t/\hbar} e^{-i\gamma_{\pm 1}} H_{MW,y,lab} e^{i\gamma_0} e^{-iH_{lab}t/\hbar} | 0, t \rangle_{lab} \\ &= \pm \frac{1}{4i} g \mu_B B_{MW} \sin \theta' e^{i(-\omega_{MW} + D \pm g \mu_B B \cos \theta \pm \omega_r \cos \theta)t} {}_{lab}\langle \pm 1, 0 | e^{i\theta S_y} S_{\pm} e^{-i\theta S_y} | 0, 0 \rangle_{lab} . \end{aligned}$$

For the transition from $|m = 0\rangle$ to $|m = \pm 1\rangle$ induced by the longitudinal microwave magnetic field, the frequency shift is $\mp \omega_r \cos \theta$, which is attributed to the Berry phase. For the transverse component, the frequency shift of $\pm \omega_r (1 - \cos \theta)$ is induced by the combination of the Berry phase and the rotational Doppler effect. In our experiment, the angle θ' of the microwave direction relative to the z axis is approximately 8.5° . Consequently, the dominant transition probability arises from the longitudinal component of the microwave, characterized by a frequency shift of $\mp \omega_r \cos \theta$ due to the Berry phase.

(iii) In Fig. 3(f) of the revised manuscript, the experimental data points (blue circles) depicting the frequency shift do not precisely align with the theoretical result $\omega_r \cos \theta$ for a fixed θ . This deviation is attributed to the change of the angle θ for different rotation frequencies. In our

experiment, the electric dipole moment of the levitated nanodiamond is not precisely parallel to the axis of the largest or smallest moment of inertia. As the rotation frequency increases, the nanodiamond tends to rotate along its stable axis, leading to a slight change in the orientation of the NV axes. This behavior is analogous to that of a gyroscope. The orientation of the NV center can be calculated by the resonance frequency at various rotation frequencies. The magenta dashed curve is a linear fitting of the resonance frequency. The orientation of the NV center rotates by approximately 3.3° at $\omega_r = 2\pi \times 20$ MHz, compared with that at $\omega_r = 2\pi \times 0.1$ MHz.

(iv) In addition, we have measured another levitated nanodiamond rotating counterclockwise, which is opposite to that shown in Fig. 3(f). Supplementary Fig. 6(c) shows the frequency shift induced by the Berry phase in a levitated nanodiamond rotating counterclockwise. The resonance frequency between the $|m = 0\rangle$ state and $|m = \pm 1\rangle$ state decreases with an increase in the rotation frequency, in contrast to the behavior observed in the levitated nanodiamond rotating clockwise. The red curve is the theoretical calculation for the angle of $\theta = 21.5^\circ$ between the NV axis and the rotating axis. The experimental data is in excellent agreement with the theoretical calculation, suggesting a constant angle θ of the NV centers at various rotation frequencies for this particular nanodiamond.

Comment 5) The authors describe the measurement in fig 2e as detecting the 'pseudo field' via a broadening of the ODMR features, while in Fig 3b they add a real magnetic field that splits the NV orientation classes so they can then resolve individual transitions and hence make a measurement of the frequency shifts imparted by rotation. This is confusing, since the measurements are essentially identical except with the addition of a bias magnetic field, and the paper reads as if the frequency shifts accumulated in the zero field case are somehow different to that when an applied field is considered (i.e. broadening vs splitting), whereas it's exactly the same and in the former they can't resolve individual transitions. I do not see the point of the zero field data when they then do the applied field experiment, which is much cleaner and more easily interpreted. These could be combined into the same figure, or the zero-field data omitted (and Eqs 1, 2, 3 revised). The significance of Eq 2 isn't well argued either, since the raising-lowering terms (the transverse magnetic field terms) can generally be ignored, they are too small even here to result in significant NV spin state mixing.

Reply: (i) We have adjusted Figures 2 and 3. Specifically, we combined the old Fig. 2(d-e) with the old Fig. 3 into a new Figure 3 in the revised version of the manuscript.

(ii) Conducting measurements without an external magnetic field (old Fig. 2(d-e)) is essential to isolate and demonstrate the specific impact of the Berry phase. In an external magnetic field (B), the frequency shift is influenced by two components. Besides the effect of the Berry phase, there is a contribution from the changes in orientation of the NV axes, which is particularly sensitive to the angle between the direction of the magnetic field and the NV axes. On the other hand, in the absence of an external magnetic field, the observed frequency shift is only induced by the Berry

phase.

(iii) We have revised equations 1-3 as discussed in our reply to comment 3. In addition, we added a discussion of ignoring off-diagonal terms (the transverse magnetic field terms) in the revised supplementary material. In the rotating frame, the Hamiltonian of the NV center can be written as (Supplementary Eq. 21)

$$H_{rot} = UH_{lab}U^\dagger + i\partial_t U U^\dagger = e^{i\theta S_y} e^{i\phi S_z} H_{lab} e^{-i\phi S_z} e^{-i\theta S_y} + i\partial_t e^{i\theta S_y} e^{i\phi S_z} e^{-i\phi S_z} e^{-i\theta S_y}$$

$$= \hbar \begin{pmatrix} D + g\mu_B B \cos \theta & -\frac{g\mu_B B \sin \theta}{\sqrt{2}} & 0 \\ -\frac{g\mu_B B \sin \theta}{\sqrt{2}} & 0 & -\frac{g\mu_B B \sin \theta}{\sqrt{2}} \\ 0 & -\frac{g\mu_B B \sin \theta}{\sqrt{2}} & D - g\mu_B B \cos \theta \end{pmatrix} + \hbar \begin{pmatrix} -\omega_r \cos \theta & \frac{\omega_r \sin \theta}{\sqrt{2}} & 0 \\ \frac{\omega_r \sin \theta}{\sqrt{2}} & 0 & \frac{\omega_r \sin \theta}{\sqrt{2}} \\ 0 & \frac{\omega_r \sin \theta}{\sqrt{2}} & \omega_r \cos \theta \end{pmatrix}$$

The second term on the right side of the equation represents the Zeeman interaction arising from the pseudo-magnetic field due to the rotation of the NV center. In the case of an adiabatic process, $\omega_r \ll D - g\mu_B B \cos \theta$, the second term is significantly weaker than the first term and can be treated as a perturbation. Moreover, the off-diagonal terms can be ignored since $g\mu_B B \ll D$, which are too small to induce significant mixing of the NV spin states. Thus, the Hamiltonian can be approximated as (Supplementary Eq. 32)

$$H_{r,B} = \hbar \begin{pmatrix} D + g\mu_B B \cos \theta - \omega \cos \theta & 0 & 0 \\ 0 & 0 & 0 \\ 0 & 0 & D - g\mu_B B \cos \theta + \omega \cos \theta \end{pmatrix}$$

Comment 6) The analysis for the data presented in Fig 2(f) and Fig 3(c) seems very simple to the point of trivial and thus not very informative at all. For Fig 2f, why are two (random) angles (0, 20deg) chosen, when the 4 orientation classes of NVs in the diamond would each make a separate angle to the rotation axis? The authors say in the manuscript that the broadening is determined mainly by NVs closer to the rotation axis, but I can't see why the authors couldn't use the inferred orientations deduced from the magnetic field data later? For Fig 3e A quantity of interest such as the rotation frequency when the angle changes could be deduced without much effort. Perhaps a piecewise linear fit?

Reply: (i) Two different levitated nanodiamond particles were employed in the measurements with an external magnetic field (new Fig. 3(f)) and without an external magnetic field (new Fig. 3(c)), so we cannot use the NV orientation inferred from data with a magnetic field to explain the data without a magnetic field. The change in diamond particles occurred because the one used in the measurement of no magnetic field was lost during the installation of magnets. As a result, the orientations of NVs in the two measurements are different. We have added a sentence in the main text to clarify this: “Data shown in Fig. 3(e),(f) are taken from one levitated diamond, which is different from the one used for Fig. 3(b),(c).”

(ii) We have added a linear fit (magenta dashed curve) of the resonance frequency in Fig. 3(f) of

the revised manuscript. The orientation of the NV center can be calculated by the resonance frequency at the various rotation frequencies. The orientation of the NV center changes by approximately 3.3° at $\omega_r = 2\pi \times 10$ MHz, compared with that at $\omega_r = 2\pi \times 0.1$ MHz. In addition, we measured another levitated nanodiamond rotating counterclockwise (opposite to the one shown in Fig. 3(f)) and added the result in Supplementary Fig. 6(c). For this new diamond particle, the experimental data is in excellent agreement with the theoretical calculation for the angle of $\theta = 21.5^\circ$, suggesting a constant angle θ of the NV centers at various rotation frequencies.

Comment 7) Key details regarding the diamond sample and experiment are entirely omitted, and these undermine the claims of the paper significantly. For example while the authors mention T1 as being 3.6ms, this is not the time that matters for coherent quantum control: what is T2, T2*? The supplement describes a measurement of "T2rabi", which is not a very meaningful metric of the spin coherence of the NVs (it says just as much about the stability of the mw field as the inhomogeneous broadening of the NVs). The authors claim that this work spins diamonds "about three orders of magnitudes faster than prior achievements using diamonds mounted on motor spindles ... this rotation speed surpasses the dephasing rate of NV spins in the diamond" but that has not been shown until we know the T2, T2* of the NVs in this diamond. In particular, the demonstration of Rabi oscillations was only possible when the rotation speed was reduced to 100kHz, so T2, T2* would need to be 10us, which is rarely the case in nano-diamonds. Additionally, it is unclear if all the measurements were taken on one nano-diamond (impressive if the case) or if many nano-diamonds were sacrificed in obtaining these results. This is important in the context of reproducibility. For example the change in orientation angle as a function of rotation I would imagine would be different for each nano-diamond.

Reply: (i) We have provided more details of the diamond sample in the supplementary information. The diamond particles utilized in this study were purchased from Adamas Nano. The product model is MDNV1umHi10mg (1 micron Carboxylated Red Fluorescence, 1 mg/mL in DI Water, ~ 3.5 ppm NV). These particles have an average size of 750 nm. They are created by irradiating 2-3 MeV electrons on diamonds manufactured by static high-pressure, high-temperature (HPHT) synthesis and containing about 100 ppm of substitutional N.

(ii) The T_2^* and T_2 measurements of a levitated nanodiamond have been added, as shown in supplementary Fig. 7(c) and Fig. 7(d) in the updated supplementary information. The T_2^* and T_2 are 40 ns and 0.52 μ s, respectively. The oscillation in the spin echo measurement is induced by the misalignment of the magnetic field with the rotation axis. Moreover, the T_2^* and T_2 measurements of a nanodiamond fixed on a substrate are shown in supplementary Fig. 8(e) and Fig. 8(f). The T_2 is 2.98 μ s, and the T_2^* is 80 ns for a fixed nanodiamond. Thus the maximum rotation speed of a levitated diamond achieved in this experiment (20 MHz) surpasses the typical dephasing rate of NV spins in the diamond.

(iii) The quantum control (Rabi) of NV centers is measured within one rotation period of the

levitated diamond particle. Fig. 4(b) shows the pulse sequence of the Rabi oscillation measurement. All the Rabi measurements in Fig. 4 are carried out at a rotation frequency of 100 kHz. The rotation period is 10 μ s which is much longer than the microwave pulse ($\leq 1 \mu$ s). The time gap (20 μ s) between the initialization and the readout laser pulses is twice that of the rotation period, which allows us to apply the microwave pulse at an arbitrary rotation phase between 0 and 2π . Therefore, the Rabi measurements do not require that the T_2 or T_2^* to be longer than the rotation period of the levitated diamond particle.

(iv) In the main text of the manuscript, the experimental data are obtained from four different diamond particles. We have added a new paragraph about diamond particles in the Method section: “The diamond particles were acquired from Adamas Nano. The product model is MDNV1umHi10mg (1 micron Carboxylated Red Fluorescence, 1 mg/mL in DI Water, \sim 3.5 ppm NV). The experimental data shown in the main text of the manuscript are obtained from four different diamond particles. The data presented in Fig. 1, Fig. 2, and Figs. 3(a-c) originate from measurements conducted on the same nanodiamond particle. Figs. 3(d-f) show the data from a second nanodiamond particle, while the data in Fig. 4 is measured using the third nanodiamond particle. Fig. 5 uses the fourth diamond particle.”

The change in the orientation angle as a function of rotation speed is different for each nanodiamond. Supplementary Fig. 6(c) displays the frequency shift induced by the Berry phase of the third nanodiamond, rotating counterclockwise, which is opposite to the one shown in Figs. 3(f). For the third nanodiamond, the experimental data is in excellent agreement with the theoretical calculation for the angle of $\theta = 21.5^\circ$, suggesting a constant orientation of the NV centers at various rotation frequencies.

Comment 8) The angular variation of the Rabi frequency due to the angle of the microwave field to the rotation axis is a well-characterised effect that does not change with rotation speed, the authors should consult and cite A. A. Wood et al Physical review letters 124 (2), 020401 2020, where the angular variation of the effective microwave phase is also discussed in detail, as well as in A. A. Wood et al Physical Review Research 3 (4), 043174 2021 where angular variation of the microwave and rf rabi frequency is characterised and corrected for.

Reply: We have added the citation of two mentioned references: [61] Physical review letters 124, 020401 (2020); [62] Physical Review Research 3, 043174 (2021).

9) The discussion and outlook of the work is quite disappointing. There is no discussion of the limitations in the experiment that need to be overcome or subject to further work, eg. the consequences of Rabi frequencies on par with the rotation speed, the tradeoff between green laser intensity and NV spin preparation time (1ms at 100kHz says there's a lot more work to do). Are higher rotation speeds possible? What are the effects of the rotation on the ^{14}N hyperfine interaction, or ^{13}C spins in the diamond (which at 20MHz see a magnetic pseudo field of almost 2T and should start to polarise, creating a real Barnett field!). The authors might also like to

consider how their results impact the other applications rapidly rotating NV centres enables, eg. improved magnetometry A. A Wood et al Physical Review Applied 18 (5), 054019 2022. Regarding NV based gyro sensors, the actual demonstrated work of Soshenko et al Phys. Rev. Lett. 126, 197702 2021 and Jarmola et al Sci. Adv 7 eab13840 2021 should be cited.

Reply: (i) We have revised the discussion section. In the second paragraph of the discussion, we added two sentences: “20 MHz rotation can generate a pseudo-magnetic field of 0.71 mT for an electron spin, and a pseudo-magnetic field of 6.5 T for an ¹⁴N nuclear spin.” “Using feedback cooling, the CoM of the levitated nanodiamond is cooled in all three directions with a minimum temperature of about 1.2 K along one direction.” We also added a new paragraph in the discussion: “The maximum rotation frequency in this experiment is limited by the bandwidth of the multichannel waveform generation system for generating the phase-shifted signals on the four electrodes. The rotation frequency can be much higher with a better waveform generation system. Furthermore, in the presence of a DC external magnetic field, the NV centers within a rotating nanodiamond experience an AC magnetic field. Quantum sensing of an AC magnetic field can have a higher sensitivity compared to that of a DC magnetic field [63]. Consequently, the mechanical rotation can enhance the sensitivity of a magnetometer in measuring DC magnetic fields. By using purer diamond particles, i.e. CVD diamonds, a higher excitation power of the 532 nm laser can be employed to reduce the initialization time of NV centers.”

(ii) We have cited the mentioned references: [63] Physical Review Applied 18, 054019 (2022); [58] Phys. Rev. Lett. 126, 197702 (2021); [59] Sci. Adv 7, eab13840 (2021).

10) Finally, I would say there is a sense of sloppiness in the preparation of the manuscript, with many typos and errors (discussed above). The many typos are perhaps best exemplified by "Date Availability" instead of "Data Availability". This is followed by what appears to me to be a meaningless statement "All data generated and analyzed in this study are available within the article and its Supplementary Information." Does this mean the data is not available upon request?

Reply: We apologize for the oversight. We have revised the manuscript and uploaded the source data for all figures shown in the main text. The updated section of Data Availability reads: “Source data for figures in the main text are provided with this paper. Other data that support the findings of this study are available from the corresponding author upon reasonable request.”

To summarise this manuscript reports a significant experimental advance and is arguably a groundbreaking moment for the study of rotating quantum systems and levitodynamics, enabling a swath of new fundamental experiments. However, I would only recommend publication after ALL of the above 10 points have been fully addressed

Reply: We thank the referee for the positive evaluation of the novelty and significance of our work. We have revised the manuscript according to the referee’s comments and suggestions.

Reply to Reviewer #2:

Jin et al levitate nanodiamonds using a surface ion trap and take nanodiamonds to high vacuum. They then demonstrate nitrogen-vacancy ODMR, fast rotation of the nanodiamond, and measure the spin coherence time. This is indeed the first demonstration of ODMR at high vacuum which has been an important obstacle. This is of wide interest to the specific growing community of levitated nanoparticles as well as the NV community. In addition, it will be of interest to the wider quantum technology community. The work supports its conclusions and claims. The methodology is sound. Overall, it would make a good paper for Nature Communications if the following questions can be answered satisfactorily:

Jin et al levitate nanodiamonds using a surface ion trap and take them to high vacuum. They then demonstrate nitrogen-vacancy ODMR including Rabi oscillations and fast rotation of the nanodiamond. This is indeed the first demonstration of ODMR of a levitating particle at high vacuum which has been an important obstacle. The extra things are excellent also: measurements where the diamond is spinning faster than the NV spin decoherence is a new regime: 1000 times faster spinning than previous NV ODMR. The Rabi oscillations are the first quantum control of spins while levitated in high vacuum. This work is of great importance to the specific fast-growing community of levitated nanoparticles, which is working on applications in future quantum sensors and tests of fundamental physics, such as for studying the quantum nature of gravity. The NV community will want to know about this work also. In addition, it will be of interest to the wider quantum technology community. The experimental work supports the conclusions and claims. The methodology is sound. Overall, it would make a great paper for Nature Communications if the following questions can be answered satisfactorily:

Reply: We extend our appreciation to the Referees for their positive evaluation of the novelty and significance of our work. We have revised the manuscript according to their suggestions.

Comment 1) The NV ODMR linewidth (~ 30 MHz) of the levitated diamonds in figure 1e and 2e is large compared to the established literature and compared to the measurements with the nanodiamonds on a substrate (supplementary info figure 8a). In figure 4c the ODMR looks sharper and more like a simple NV ODMR spectrum. Some explanation of this should be given in the manuscript.

Reply: (i) Thanks for the Referees' comments. We have added an explanation for the broad linewidth of ODMR in the revised manuscript. In Fig. 1(e), the Lorentzian fitting of the ODMRs shows a linewidth of approximately $2\pi \times 19$ MHz for both the blue and red curves, with a strain effect splitting E of $2\pi \times 6.7$ MHz. The blue curve in Fig. 3(b) (the old Fig. 2(e)) has the same

linewidth as that in Fig. 1(e). The Full Width at Half Maximum (FWHM) of red curve in Fig. 3(b) is much boarder because of the effect of the Berry phase in rotating NV centers. In contrast, in Supplementary Fig. 8(a), the ODMR linewidth ($2\pi \times 7.4$ MHz) is narrower as we measured it with a lower microwave power and the energy levels of the four orientation NV centers are separated. These ODMR linewidths are common for nanodiamond particles.

(ii) In Fig. 3(e), the ODMR is measured with an external magnetic field. Thus, the energy levels of NV centers with different orientations are separated. The linewidth of levitated diamond NV centers is larger than that of fixed diamond NV centers due to the continuous change of the NV orientations in levitated diamonds relative to the magnetic field.

Comment 2) The manuscript should describe the type of nanodiamond used e.g. is it purchased, isotopically purified, HPHT or CVD grown etc.

Reply: We have added this information in the Method section and the supplementary information. The diamond particles utilized in this study were purchased from Adamas Nano. The product model is MDNV1umHi10mg (1 micron Carboxylated Red Fluorescence, 1 mg/mL in DI Water, ~ 3.5 ppm NV). These particles have an average size of 750 nm. They are created by irradiating 2-3 MeV electrons on diamonds manufactured by static high-pressure, high-temperature (HPHT) synthesis and containing about 100 ppm of substitutional N.

Comment 3) It would be interesting to cool the centre-of-mass (CM) motion of the nanodiamonds. However, the authors do not do so despite the fact that they have significant expertise in this field. Is there some reason for this?

Reply: We have performed an additional experiment to cool the center-of-mass motion of a levitated nanodiamond. In the revised main text of the new manuscript, we have introduced a new section and a new figure 5 discussing the feedback cooling of the center-of-mass (CoM) motion. The schematic diagram is shown in Fig. 5(a). The temperatures of the CoM motion in the x, y, and z directions are cooled to about 1.2 K, 3.6K and 86 K, respectively, as shown in Fig. 5(b)-(d). Compared with the temperatures of the CoM motion in the x and y directions, the poor cooling effect in the z direction is caused by the low signal-to-noise ratio, limited by the small center hole of the surface ion trap.

Comment 4) The lowest pressure the authors have achieved is $6.9e-6$ Torr. Is there any limitation here for why the authors didn't go further down in pressure?

Reply: In our experiment, the lowest pressure is primarily constrained by the outgassing of the vacuum chamber, the viton gaskets and the speed of the vacuum pump. The variation in internal temperature of a levitated diamond at different pressures is shown in Fig. 1(f). The temperature stabilizes at approximately 350 K when the pressure is below $5e-5$ Torr. So, the lowest pressure

that we have achieved is already low enough to show this. Ultra-high vacuum can be achieved in the future with a better vacuum system and bakeout.

Comment 5) Some further comments to improve the manuscript:

A. It should be mentioned that the Stark shift for NV would be small and so is not relevant here.

Reply: Thanks for the Referee's comments. We have added the note in the third paragraph of the section "Fast rotation and Berry phase" in main text: "The Stark shift for NV centers induced by the electric field is negligible and hence is not included in the equation."

Comment 6) B. In Figure 1e there is a splitting due to the E term coming from strain in the nanodiamonds. This point should be explained for non-experts.

Reply: We have included an explanation for the splitting observed in the ODMR in the absence of an external magnetic field in the method section: "In the experiment, we measure the ODMR of levitated nanodiamond NV centers to detect the internal temperature in the absence of an external magnetic field. The zero-field Hamiltonian of NV center is: $H = DS_z^2 + E(S_x^2 - S_y^2)$, where D is the zero-field energy splitting between the $|m = 0\rangle$ state and $|m = \pm 1\rangle$ state, E is the splitting between the states due to the strain effect. The small splitting between two dips in the ODMR spectra (Fig. 1(e)) without an external magnetic field is due to the E term from strain in the nanodiamond."

Comment 7) C. In Figure 2d the caption should say that the nanodiamond is rotating about the z axis, not along the z axis.

Reply: We have revised this. The words "along the z axis" have been changed to "around the z axis."

ATM Anishur Rahman and Gavin W Morley

Reply to Reviewer #3:

I co-reviewed this manuscript with one of the reviewers who provided the listed reports. This is part of the Nature Communications initiative to facilitate training in peer review and to provide appropriate recognition for Early Career Researchers who co-review manuscripts

Reply: We thank the referee for the assessments and valuable comments. We have revised the manuscript accordingly.

Reply to Reviewer #5:

Recommendation: This paper is publishable subject to minor revisions noted. Further review is optional.

Comments:

This paper presents exciting results on the quantum control and fast rotation of levitated nanodiamonds (NDs) containing NV centers in high vacuum. The authors developed a planar Paul trap with an integrated microwave antenna to trap nanodiamonds down to 10^{-5} Torr. They show that they can rotate the NDs up to 20MHz, faster than the dephasing rate of the NVs. They take advantage of the NVs to estimate the particle's internal temperature and probe the pseudo magnetic field generated by the rotation, the Barnett effect. Finally, they show coherent control of the NVs while the ND is rotated at 0.1MHz. This work sets a new milestone for the levitated optomechanics community. It shows a promising technical solution for applied and fundamental physics such as gyroscope, the generation of massive quantum superposition using spins or the detection of quantum gravity. I recommend this article for publication in Nature Communication, provided that the authors address my questions.

Reply: We express our appreciation to the Referee for the positive evaluation of the novelty and significance of our work, as well as the recommendation for publication. We have revised our manuscript to address the Referee's questions and comments.

(1) In the abstract, the authors claim that “fast rotation of levitated diamond has not been reported.” This statement should be removed as it is not quantitative. What is considered fast here?

Reply: (i) We have removed the sentence “In addition, fast rotation of a levitated diamond has not been reported” in the abstract.

(ii) In the experiment, we drove a nanodiamond to rotate up to 20 MHz, surpassing typical NV electron spin dephasing rates. This is about 3 orders of magnitudes faster than previous achievements using diamonds mounted on electric motor spindles.

(2) Introduction, paragraph 2, regarding the levitation of diamonds at low vacuum level, the author should add the following reference: M. C. O'Brien & al., Appl. Phys. Lett. 114, 053103 (2019)

Reply: We have added the citation of the mentioned reference: [49] Appl. Phys. Lett. 114, 053103 (2019).

(3) Introduction, paragraph 3, line 3, what did the authors mean by “it compromises multiple electrodes...”?

Reply: We apologize for the typo. We have changed the words from “it compromises multiple electrodes...” to “it comprises multiple electrodes...”.

(4) Introduction, paragraph 3, second to last phrase. I suggest changing the phrase “we achieve the quantum coherent control ...”. Is the control coherent or quantum? Can it be quantum without being coherent and vice-versa?

Reply: We have removed the word “coherence” in the mentioned phrase. So it now reads “...we achieve quantum control of NV centers...”

(5) In Figure 1, I need help finding the method used for the electric field simulation in the supplementary. Add a phrase or two on the method used, at least in the supplementary.

Reply: The electric field of the surface ion trap shown in Fig. 1(c) and the rotating electric field used to drive levitated diamonds to rotate are both simulated by the COMSOL software. We have added the description in the Supplementary Information: “We simulate the electric fields for both the trapping potential and the rotating driving field using the COMSOL software.”

(6) Part C, first phrase: “Quantum coherent control...”, same remark as in (4), I would suggest using either quantum or coherent.

Reply: We have modified the sentence from “Quantum coherent control of spins are important...” to “Quantum control of spins are important...”.

(7) Fig 4. Title, same remark as above.

Reply: We have modified the sentence from “Quantum coherent control of NV centers in a levitated nanodiamond...” to “Quantum control of NV centers in a levitated nanodiamond...”.

(8) In Figure 4e, I suggest using a different color scheme than the one in Fig. 4d to avoid confusing readers. Which transitions (1, 2 or 3) correspond to $\theta=22$ degree? Make it clear in the legend and caption.

Reply: (i) In Fig. 4(e), we have changed the color of the Rabi oscillation curve measured at rotation phase of π from red to black. The Rabi oscillation measured at rotation phase of $\pi/2$ (blue curve)

is same with the blue curve in Fig.4(d)

(ii) The Rabi oscillations in Fig. 4(e) are measured at the transition frequency of 3.129 GHz (dip 3), which corresponds to the NV center orientation of 22°. We have labeled this in the Fig. 4(e) and 4(f). In the caption of Fig. 4(e), we have added the note: “Rabi oscillation of NV centers with $\theta = 22^\circ$, corresponding to the resonance frequency of 3.129 GHz (dip 3)”.

(9) Part C, formula (4) It would make the article and statement more straightforward if the authors added a small discussion (one phrase or two) on why the Rabi frequency is affected by the drive phase phi. The authors mentioned it in the supplementary, but moving it to the main text will be better.

Reply: We have added the corresponding discussion in the section of “Quantum control of fast rotating NV centers” in the main text. “Due to the Ω -shape of the microwave antenna, the orientation of the magnetic field of the microwave is located in yz-plane and slightly different from the z axis with an angle of about $\theta' = 8.5^\circ$ (Fig. 4(a)). So, $n_{MW} = (-\sin\theta', 0, \cos\theta')$. The effective microwave magnetic field acting on NV spins, with the orientation of $n_{NV} = (\cos\phi(t)\sin\theta, \sin\phi(t)\sin\theta, \cos\theta)$, changes as a function of the rotation phase $\phi(t)$ of the levitated nanodiamond. Therefore, the Rabi oscillation frequency Ω_{Rabi} can be written as [61, 62]:

$$\Omega_{Rabi} \propto \sqrt{1 - (\cos\theta\cos\theta' - \sin\phi(t)\sin\theta\sin\theta')^2} \text{ ,”}$$

(10) The authors should add a reference to the following article as it is very much related to the authors’ article: M. Perdriat & al., “Spin Read-out of the Motion of Levitated Electrically Rotated Diamonds”, arXiv:2309.01545 (2023)

Reply: We have cited the mentioned reference: [47] arXiv:2309.01545 (2023).

REVIEWERS' COMMENTS

Reviewer #1 (Remarks to the Author):

The authors have addressed all of the concerns raised in the original report (comprehensively) and as such I recommend publication of the revised manuscript subject to one minor change:

Change

"DATE AVAILABILITY" to "DATA AVAILABILITY"

Reviewer #2 (Remarks to the Author):

The changes the authors have made fully answer our questions, including adding in nice new work on cooling the COM motion. We recommend publication now with no further changes requested.

Reviewer #3 (Remarks to the Author):

I co-reviewed this manuscript with one of the reviewers who provided the listed reports. This is part of the Nature Communications initiative to facilitate training in peer review and to provide appropriate recognition for Early Career Researchers who co-review manuscripts."

Reviewer #5 (Remarks to the Author):

The authors addressed my questions so I would like to recommend this article for publication in Nature Communication.

Point-by-point responses to reviewers' reports

Reviewer #1 (Remarks to the Author):

The authors have addressed all of the concerns raised in the original report (comprehensively) and as such I recommend publication of the revised manuscript subject to one minor change:

Change

"DATE AVAILABILITY" to "DATA AVAILABILITY"

Our reply: We thank the reviewer for recommending our manuscript for publication. We have changed "DATE AVAILABILITY" to "DATA AVAILABILITY."

Reviewer #2 (Remarks to the Author):

The changes the authors have made fully answer our questions, including adding in nice new work on cooling the COM motion. We recommend publication now with no further changes requested.

Our reply: We thank the reviewer for recommending our manuscript for publication.

Reviewer #3 (Remarks to the Author):

I co-reviewed this manuscript with one of the reviewers who provided the listed reports. This is part of the Nature Communications initiative to facilitate training in peer review and to provide appropriate recognition for Early Career Researchers who co-review manuscripts."

Our reply: We thank the reviewer for recommending our manuscript for publication.

Reviewer #5 (Remarks to the Author):

The authors addressed my questions so I would like to recommend this article for publication in Nature Communication.

Our reply: We thank the reviewer for recommending our manuscript for publication.